# Ratiometric detection of perfluoroalkyl carboxylic acids using dual fluorescent nanoparticles and a miniaturised microfluidic platform

Yijuan Sun [1,2], Víctor Pérez-Padilla [1,2], Virginia Valderrey[1], Jérémy Bell [1], Kornelia Gawlitza [1] & Knut Rurack [1] ✉

The widespread contamination of soil and water with perfluoroalkyl substances (PFAS) has caused considerable societal and scientific concern. Legislative measures and an increased need for remediation require effective on-site analytical methods for PFAS management. Here we report on the development of a green-fluorescent guanidine-BODIPY indicator monomer incorporated into a molecularly imprinted polymer (MIP) for the selective detection of perfluorooctanoic acid (PFOA). Complexation of PFOA by the indicator, which is mediated by concerted protonation-induced ion pairing-assisted hydrogen bonding, significantly enhances fluorescence in polar organic solvents. The MIP forms as a thin layer on silica nanoparticles doped with tris(-bipyridine)ruthenium(II) chloride, which provides an orange emission signal as internal reference, resulting in low measurement uncertainties. Using a liquid-liquid extraction protocol, this assay enables the direct detection of PFOA in environmental water samples and achieves a detection limit of 0.11 μM. Integration into an opto-microfluidic system enables a compact and user-friendly system for detecting PFOA in less than 15 minutes.

Per- and polyfluoroalkyl substances (PFAS) are a family of chemicals with fully or partly fluorinated carbon backbones. PFAS, also known as 'forever chemicals', can be categorised according to their non-fluorochemical functionalities[1,2]. If the functional group is a carboxylic acid, they are referred to as perfluorocarboxylic acids (PFCAs), which is often used as a synonym for perfluoroalkyl carboxylic acids. Among PFAS, PFCAs have gained societal and scientific attention and importance, especially because of perfluorooctanoic acid (PFOA)[3,4]. The extent of PFOA pollution is due to the intensive use of PFOA as an emulsifier for the emulsion polymerisation of Teflon in recent decades, which in turn led to its uncontrolled release into the environment on a tonne scale[3].

The extent of PFAS pollution is illustrated by their ubiquitous presence in soil and water, where they persist and bioaccumulate[5,6]. PFAS are found in virtually all ecosystems and accumulate in living organisms, including livestock and humans[7,8]. As the toxicity of PFAS has been demonstrated in a number of studies[9,10], these compounds pose a serious threat that has brought them to the attention of regulatory authorities worldwide[11,12] and has led to their classification as persistent organic pollutants (POPs) by including PFOA, PFOA-related compounds and their salts in the Stockholm Convention in 2019 to prohibit their use[13]. Therefore, there is tremendous interest in detecting PFCAs in surface, waste, agricultural, ground and drinking water[14–16]. A recent governmental study found that nearly half of all US

[1]Bundesanstalt für Materialforschung und -prüfung (BAM), Berlin, Germany. [2]These authors contributed equally: Yijuan Sun, Víctor Pérez-Padilla.
✉e-mail: knut.rurack@bam.de

tap water is contaminated with PFCAs[17]. Monitoring PFCAs as close as possible to their point of occurrence is thus an important tool to track and understand the fate of these substances in nature and to support their management[18].

However, according to the current state-of-the-art, PFAS are almost exclusively analysed using gas or liquid chromatography coupled to mass spectrometry such as GC-MS, HPLC-MS, tandem HPLC MS/MS or HRMS[19–21]. Although these techniques are very sensitive and can measure down to the ppt range, they rely on laboratory environment and trained personnel, are complicated and costly and cannot be used, for example, for samples from highly polluted industrial sites or for routine monitoring at wastewater treatment plants. There is therefore an urgent need to develop new and more cost-effective analytical methods for on-site and point-of-need analyses[22–24]. Besides electrochemical sensors[25–27], especially optical approaches are very promising in this regard[28]. The latter include refractive index or colour changes[29,30], but most approaches utilise luminescence detection as it is intrinsically more sensitive. Fluorescence-based detection of PFCAs has been demonstrated with functionalised fluorescent nanoparticles[31,32], metal-organic frameworks[33], conjugated polymers[34,35], and fluorescent dyes[36–41]. However, these approaches typically require lab-based instrumentation and suffer from drawbacks such as complexity, e.g., in multi-component detection schemes involving micelle or aggregate formation, long assay times, use of toxic materials or organic solvents. Only few approaches with on-site detection potential have been reported, using thin films, glass chips or cuvettes with smartphones and 3D-printed holders[29,30,35,38]. Furthermore, it has not yet been possible to obtain good-binding antibodies against PFOA or other PFAS, so that the development of immuno-chemical sensors has also not yet been successful.

With a chemical class as diverse as PFAS, the development of sensors is a challenge, because on-site methods have to focus on sum parameters, subclass selectivity or a small number of relevant lead compounds[28]. While fluorine as a sum parameter for PFAS is difficult to realise in a miniaturised fashion, given the strength of the C−F bond and the many different (also inorganic) fluorine sources besides PFAS in an environmental sample, and because possible lead compounds are chemically very similar aggravating simultaneous detection in a miniaturised setup, subclass selectivity seemed to be the most promising starting point. Furthermore, a viable sensing approach should at best be modular in the sense that it can be adapted for other analyte subclasses and/or customised for a range of different scenarios. Therefore, we developed a sensor system for medium- and long-chain PFCAs, focussing on the key representative PFOA. Our approach (Fig. 1) is based on

(i) a newly designed fluorescent indicator, a boron−dipyrromethene (BODIPY) dye, which carries a guanidine receptor,

(ii) operates through a concerted protonation-mediated ion pairing-assisted hydrogen bonding recognition and signalling process, the hydrogen bonds imparting directionality, the ionic interaction non-covalent binding strength and both together a strong fluorescence signal change, and

(iii) is covalently embedded in a polymeric recognition matrix, which is grown as a thin molecularly imprinted polymer (MIP) layer against PFOA on the surface of

(iv) silica core nanoparticles, which in turn contain a dye with a second fluorescence colour as a reference signal and are particularly suited for miniaturised analytical devices because of straightforward handling, largely avoiding sedimentation or clogging.

These dual fluorescent core-shell nanoparticle probes are integrated into

(v) a miniaturised microfluidic device using laser diodes for excitation and a USB spectrometer-based detection unit in which

(vi) they bind and indicate the analytes of interest after these have been extracted from an aqueous sample in

(vii) a simple phase-transfer shaking step, the microfluidic approach only requiring small amounts of an organic solvent.

The combination of these modules promised a sensitive response via an analyte-modulated green BODIPY fluorescence that is internally referenced against an orange fluorescence signal of the core, contributing to robustness and reliability. Interference from competitors or matrix effects are reduced due to a molecularly imprinted recognition layer as well as an extraction step, while the overall miniaturised assay is simple and comparatively rapid to perform, as well as cost-effective to manufacture or adapt to a wide range of on-site testing scenarios in the future.

## Results

Before presenting and discussing the development of the single modules of our approach, it is important to mention three application-oriented aspects that predetermined our design considerations.

(i) The wavelength range of operation should basically be the visible region. This guarantees that all spectroscopic instrumental components are rather inexpensive, and that the synthesis of the indicator monomer is comparatively straightforward.

(ii) The ratiometric measurement should be realised with a stable signal from a dye in the particle core and an analyte-modulated signal from an indicator in the recognition layer as the shell. Such dual fluorescent systems offer several advantages, such as robust detection in challenging environments. They outperform systems that rely solely on a ratiometric response from a single indicator in the recognition layer, since indicators capable of existing in two highly emissive yet spectrally well-separated forms−both in the absence and presence of an analyte−are rare[42]. By comparing an analyte-sensitive signal with a stable reference signal, ratiometric measurements reduce the impact of external factors like light fluctuations and matrix inhomogeneities, this self-referencing approach thus leading to more reliable and accurate detection[42,43]. While most approaches combine quantum dots or carbon dots with fluorescent dyes in core-shell architectures, fully dye-based systems offer advantages like precise control over dye concentration and straightforward integration with imprinted polymers. We therefore envisioned an orange-fluorescent core with a shielded reference emitter dye and a green-fluorescent shell with a fluorescent indicator monomer, the latter responding to the analyte via a strong "OFF−ON"-type signal enhancement for optimum sensitivity.

(iii) Finally, besides assessing the possibility for monophasic detection, which would naturally be the simplest approach, we also considered a biphasic extract-&-detect assay, which requires an additional step yet brings about several advantages in terms of signal gain and matrix suppression[44,45].

### Design, synthesis and spectroscopic properties of indicator monomer 1

For the fluorescent indicator monomer to respond to PFCAs, the boron−dipyrromethene (BODIPY) chromophore was chosen as the scaffold because of its chemical versatility and remarkable spectroscopic properties, including high brightness, narrow and intense spectral bands, and excellent photostability[46–48]. For binding strongly to the strong acid PFOA ($pK_a \approx −0.5$ to $1.0$)[49,50], a guanidine moiety was incorporated into the BODIPY scaffold as a strongly basic receptor unit ($pK_b \approx 0.4$)[51]. This unit is capable of reversible protonation-deprotonation and the conversion can be monitored by alterations in absorption and emission profiles if coupled with a fluorophoric π system like BODIPY. Furthermore, the protonated form of guanidine, the guanidinium cation, can further bind to the carboxylate of PFOA

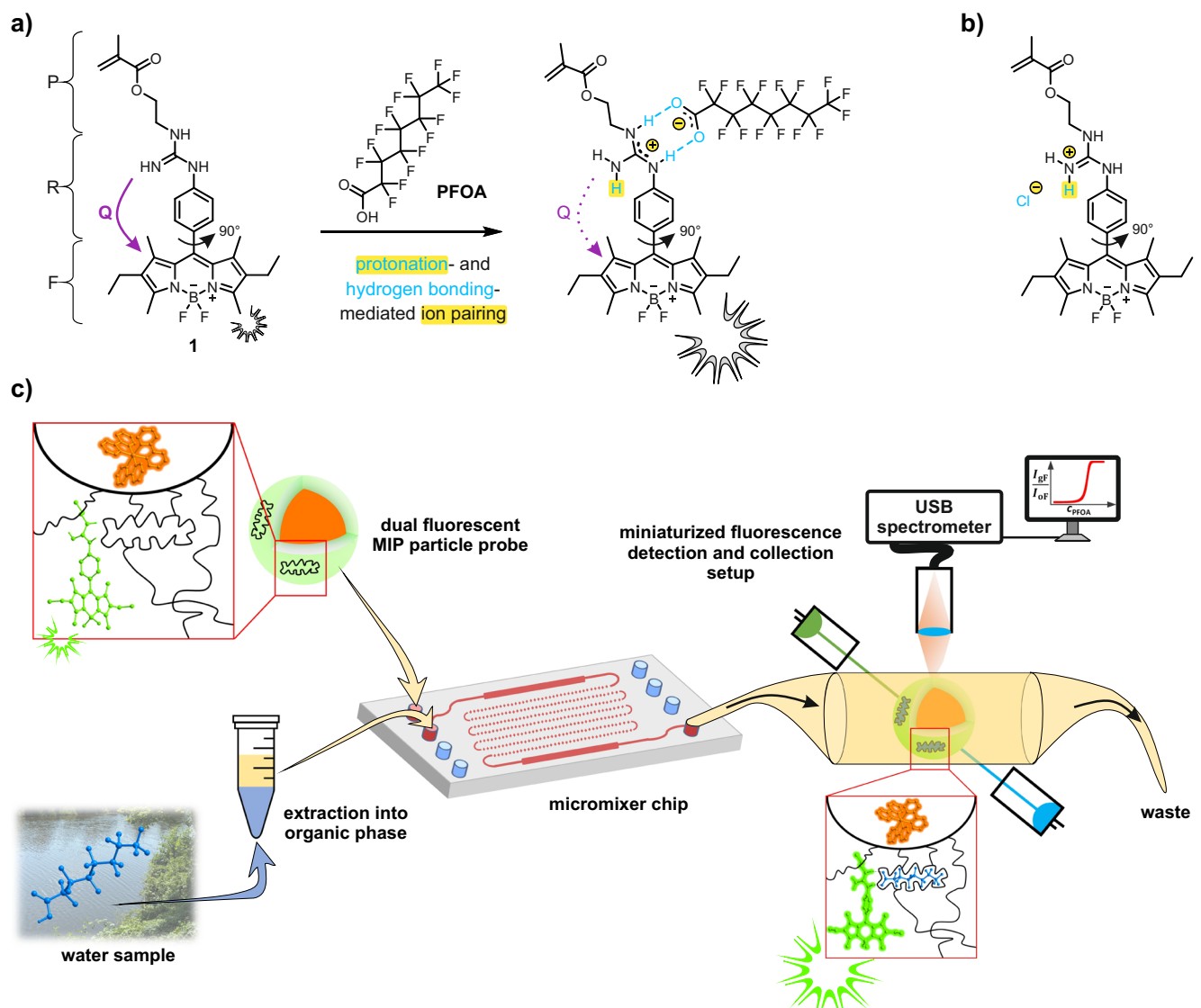

**Fig. 1 | Fluorescence switching mechanism in guanidine-BODIPY indicators and microfluidic assay workflow for PFCA detection. a** Chemical structure of the guanidine-containing BODIPY-based indicator monomer **1** (F: fluorophore, R: receptor, P: polymerizable unit) and its interaction mode with PFOA via protonation-mediated (blue and highlighted yellow) hydrogen bonding (blue) and ion pair complex formation (highlighted yellow). The active signalling process involves an electron transfer-type quenching (Q: violet arrow) from the 4-receptor-phenyl unit to the BODIPY fragment coupled through its *meso*-position, which is switched off by analyte binding, reviving the indicator's fluorescence. **b** Preferential interaction mode in HCl-protonated **1** (**1**/HCl). **c** Modular approach to a fluorescence-based microfluidic PFCA sensor (PFOA in blue). USB universal serial bus, PFOA perfluorooctanoic acid, PFCA perfluorocarboxylic acid.

through hydrogen bonding and electrostatic attraction[52], facilitating the interaction between the indicator and the analyte in a well-defined, directional manner (Fig. 1a). Directionality is important when an indicator monomer is used in combination with molecular imprinting for the generation of binding cavities in a polymer network. In analogy to BODIPYs substituted with an aniline-type receptor, a conversion between the basic (neutral) and the acidic (positively charged) state of the receptor promises to yield exceptionally high signal changes ("OFF–ON" response) when the receptor unit is introduced via the *meso*-position of the BODIPY scaffold[53] rather than the 2,6- or 3,5-positions[54,55], letting us select this arrangement for **1** (Fig. 1a). In addition to the receptor motif, **1** was functionalised with a polymerizable methacrylate moiety for subsequent integration into an analyte-responsive polymer film (Fig. 1a, c).

The synthesis of **1** started from *meso*-(4-aminophenyl)-BODIPY[56] and involved the sequential preparation of *meso*-(4-iso-thiocyanatophenyl)-BODIPY and the polymerizable thiourea-BODIPY

precursor[57], yielding **1** after treatment with ammonia solution in the presence of *N,N'*-dicyclohexyl carbodiimide as described in *Methods* below and Section II.a., Supplementary Information (65% overall yield). The structure of **1** was confirmed by [1]H NMR, [13]C NMR and high-resolution mass spectrometry (HRMS), see Figures S1–S3, Supplementary Information.

The photophysical properties of **1** were investigated in various organic solvents (Table S4, Figure S4). **1** exhibits an intense absorption at approx. 520 nm. The molar absorption coefficient, determined in ethyl acetate (EtOAc) and acetonitrile (MeCN), is comparatively high ($\varepsilon \approx 7 \times 10^4 \, \text{cm}^{-1} \, \text{M}^{-1}$) and agrees well with values obtained for other green-fluorescent BODIPY dyes[48]. The fluorescence emission spectra possess a shape that is a mirror image of the intense absorption band and are only weakly Stokes-shifted (approximately 10 nm), suggesting negligible structural changes in the excited state. Notably, the fluorescence quantum yield ($\Phi_f$) of the indicator monomer is favourably low in MeCN, tetrahydrofuran (THF) and EtOAc, i.e., $\Phi_f = 0.01, 0.02$

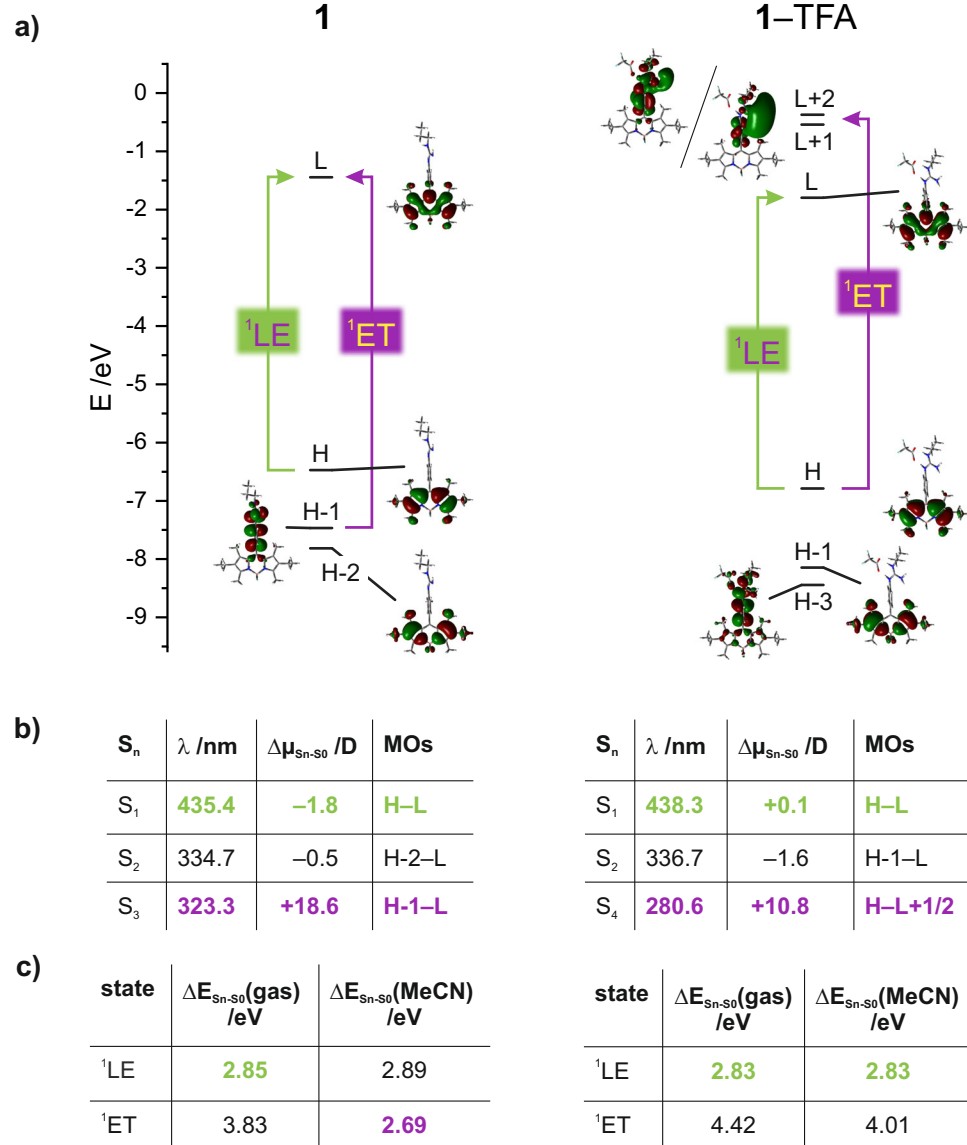

**Fig. 2 | Designed electronic features and photophysics of guanidine-BODIPY indicator. a** Energy levels and localization of relevant frontier molecular orbitals HOMO (H), HOMO−n (H − n), LUMO (L) and LUMO+n (L + n) for **1** and **1**⊂TFA (trifluoroacetic acid was used instead of PFOA for the calculations as explained in Section IV, Supplementary Information). $^1$LE and $^1$ET denote states involving transitions as indicated in the diagram and in the tables in (**b**, **c**); in (**b**) $S_n$ denotes the respective transitions at $\lambda$ as calculated in the gas phase, $\Delta\mu$ the corresponding change in dipole moment for the MOs given; **c** lists the lowest-energy emissive $^1$LE and quenching $^1$ET states in gas phase and MeCN with only the lowest, i.e., active state being highlighted; for details on calculations, see Section IV, Supplementary Information. HOMO/LUMO highest occupied/lowest unoccupied molecular orbital, TFA trifluoroacetic acid, LE locally excited state, ET electron transfer state.

and 0.03, promising a considerable fluorescence enhancement upon protonation-induced ion pairing-assisted hydrogen bonding of an analyte in the aspired analytical reaction ("OFF−ON" response). Moreover, $\Phi_f$ increases with increasing water content in MeCN/H$_2$O mixtures, reaching a maximum value of ~0.7 at H$_2$O/MeCN ≥ 80/20 (v/v), see Figure S4b. This water-dependent fluorescence enhancement is attributed to protonation of the strongly basic guanidine moiety already by an excess of water, forming the guanidinium twin. Similarly, in protic solvents and solvents that can contain/potentially form acids, a higher fluorescence is found (Table S4, Supplementary Information).

Figure 2a illustrates the supramolecular photophysical rationale behind the signalling mechanism (for a detailed description, see Section IV, Supplementary Information including Tables S5–S7 and Figure S5). In neutral **1**, the highest occupied and the lowest unoccupied molecular orbital, HOMO and LUMO, are localized on the BODIPY

fragment. In addition, the lowest-energy S$_1$ transition is a HOMO−LUMO transition localized on BODIPY (Fig. 2a, b, Table S5) and thus involves a negligible change in dipole moment upon excitation, as manifested in virtually unchanged absorption maxima across the entire solvent polarity range (Tables S4, S6), as typically associated with a locally excited singlet state ($^1$LE state; green arrows and entries in Fig. 2a, b). Whereas the S$_2$ transition is also centred on BODIPY (Table S5), involving HOMO − 2 and LUMO and showing similar features as S$_1$, the S$_3$ transition involves an electron transfer-type transition from HOMO − 1, localized on the *meso*-(*p*-guanidino)phenyl (PhGua) fragment, to the BODIPY-localized LUMO, which is connected with a large dipole moment change as is typical for electron transfer states ($^1$ET state; violet arrows and entries in Fig. 2a, b; Table S5). Accordingly, when the stabilization of this $^1$ET state by a polar solvent such as MeCN is considered (Fig. 2c, Table S6), it can become

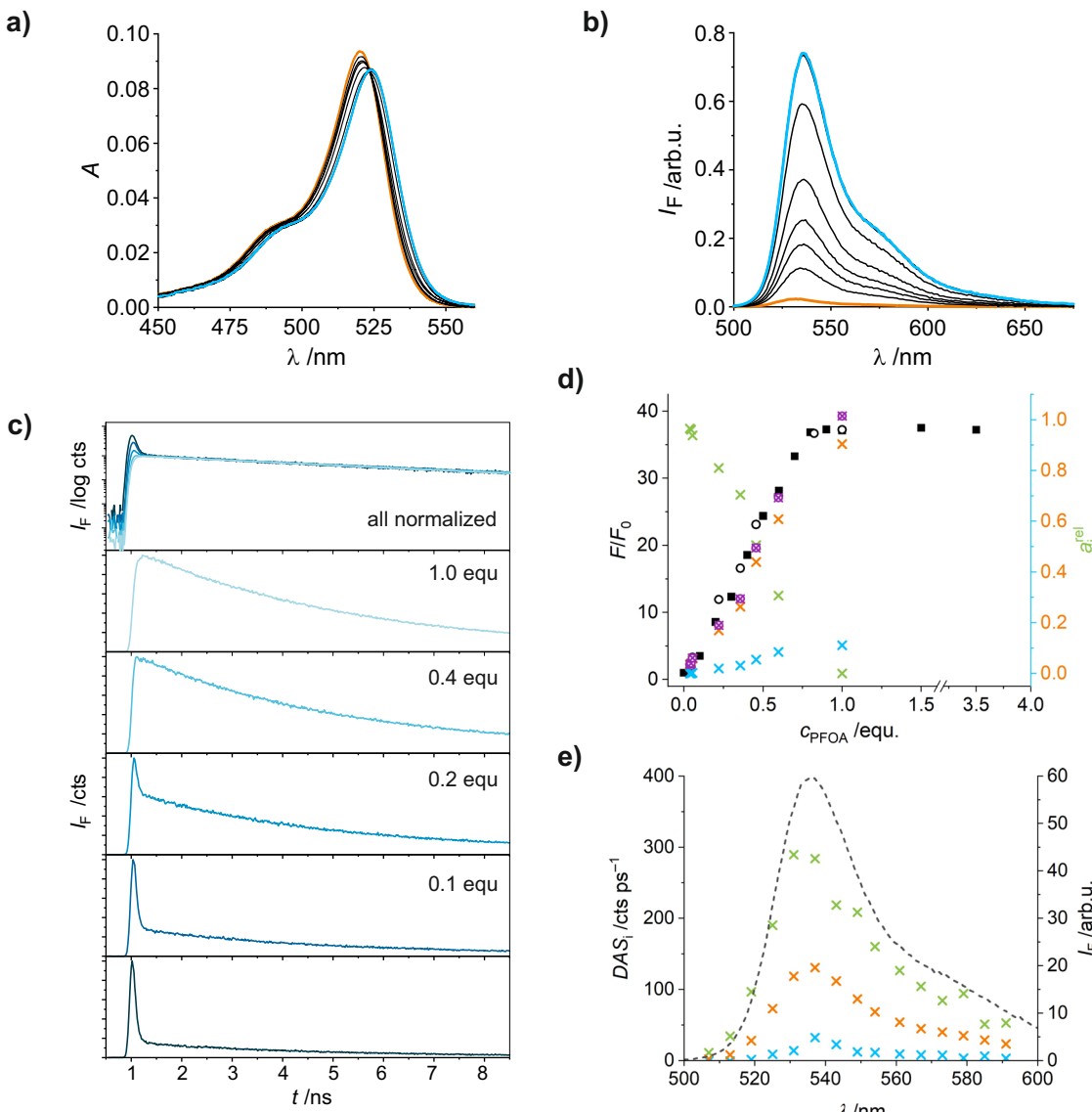

**Fig. 3 | Spectroscopic response of guanidine-BODIPY indicator upon binding of PFOA.** The titrations in (**a**, **b**) show absorption and emission spectra of **1** in the absence (orange) and the presence of increasing concentrations of PFOA according to the second titration protocol given in Section I.c.i., Supplementary Information (0.2–3.6 μM, end point spectra in blue). **c** Shows the fluorescence decays of selected titration steps as given where the top panel collects the five decays in semi-logarithmic representation after normalization on the long decay component. **d** Reduced fluorescence $F/F_0$ of two steady-state fluorescence titrations (black squares, white circles; left axis) and relative amplitudes $a_i^{rel}$ of the three decay

components (×, green for $i = 1$ [$\tau_1 = 0.03$ ns], blue for $i = 2$ [$\tau_2 = 0.51$ ns], orange for $i = 3$ [$\tau_3 = 4.35$ ns]; right axis) necessary to best describe the time-resolved emission behaviour of the system (crossed circle = $a_2^{rel} + a_3^{rel}$), see text, Sections I, VI, Supplementary Information and Equation S4 and Figure S9a for fit of titration data. **e** Decay associated spectra ($DAS_i$) of the three species (crosses, colour code as in **d**) reconstructed from the wavelength-resolved fluorescence decays of **1** in the presence of 0.3 equiv. PFOA according to Equation S8 (MeCN, $c_1 = 1$ μM, $\lambda_{exc} = 480$ nm; steady state spectrum included as dotted line for comparison).

energetically lower-lying than the BODIPY-localized $^1$LE state and lead to an effective quenching of the fluorescence as seen in MeCN, THF or EtOAc (violet entry in Fig. 2c, left table; Tables S4, S6). When now considering protonation of the guanidine unit and subsequent oxoanion binding, the electron density on the *meso*-group is reduced and the potentially quenching lowest-energy $^1$ET state (S$_4$) involving BODIPY-centred HOMO and PhGua-centred LUMO + 1 and LUMO + 2 becomes energetically unfavoured even in highly polar solvents (Fig. 2a–c, entries for $^1$ET in Fig. 2c, right table; Table S7). As a result, the fluorescence should be switched back on, despite the fact that an anion is bound to the indicator.

**Binding of indicator monomer 1 with perfluorooctanoic acid**
The interaction between **1** and PFOA was thus investigated in various solvents to select the most promising approach for assay design. In

MeCN, a polar non-protic solvent in which PFCAs are well soluble, a small bathochromic shift of the absorption and emission bands and a large fluorescence enhancement were observed upon binding of PFOA (Fig. 3a, b). In contrast, no significant interactions between **1** and PFOA were spectroscopically detectable in the polar protic solvents EtOH and MeOH (Figure S6).

Together with the fact that **1** shows distinctly higher florescence quantum yields in both alcohols compared to MeCN, see Table S4, these results suggest that the guanidinium form is already the prevalent one in protic solvents which precludes strong protonation-induced signal changes. Figure S4b further shows that not only are alcohols disadvantageous for assay development, but also MeCN/H$_2$O mixtures would only allow a narrow operating range between MeCN/ H$_2$O 99/1–95/5 (v/v) to achieve pronounced signal changes, which is impractical when aiming for a robust assay for on-site analysis. The low

tolerance of protic solvents suggests that biphasic extract-&-detect assays are preferable to monophasic ones[44,45,58], which led us to medium polar aprotic solvents that are immiscible with water. The interaction of **1** and PFOA was thus investigated in EtOAc and propyl acetate (PrOAc). As can be deduced from Figure S7a–c, a similar behaviour as for **1** in MeCN was found for both acetate solvents, i.e., small bathochromic spectral shifts and an enhancement in fluorescence upon binding of PFOA. Signal saturation differed for each solvent, following the trend MeCN (39-fold increase) > EtOAc (18-fold increase) > PrOAc (10-fold increase, Figure S7d), which agrees with the described behaviour for charge or electron transfer in substituted *meso*-BODIPYs and is supported by the theoretical data (Tables S6, S7)[59,60]. As explained above, the architecture of **1** consists of an electron-rich guanidine receptor that is directly π-conjugated via the *para*-position of a *meso*-appended phenyl ring to the BODIPY fluorophore acting as an electron acceptor, the phenyl and BODIPY π systems being perpendicularly oriented (Figs. 1a, 2a). For such virtually decoupled donor–acceptor ensembles, spectroscopic responses are enhanced with solvent polarity, regardless of whether the interaction results in an increase or a decrease. The more polar a solvent is, the better it can stabilise charge or electron transfer states, see also Fig. 2c and Table S6. As such states are commonly the OFF states of an indicator, quenching its fluorescence ("Q" in Fig. 1a), their $\Phi_f$ is lower in more polar solvents which in turn leads to stronger spectroscopic responses in the "OFF–ON" signalling process. Here, the decrease in $\Phi_f$ of **1** in the series PrOAc (0.04) > EtOAc (0.03) > MeCN (0.01) is thus mainly responsible for the trend in the PFOA-induced fluorescence enhancement observed in Figure S7d, the fluorescence quantum yields of the complex in the three solvents being rather similar with $\Phi_f = 0.45 \pm 0.08$. Based on the spectroscopic findings, the interaction of **1** with PFOA proceeds via a protonation-mediated hydrogen bonding and ion pairing reaction. The electron-rich guanidine is converted into an electron-poor guanidinium unit upon protonation through the analyte, which in turn leads to a strong, ion paired and hydrogen bonded complex as a result of electrostatic attraction between the two Y-shaped, oppositely charged H-bonding motifs of the partners. The interaction turns **1** from a weakly fluorescent guanidine state (e.g., $\Phi_f = 0.01$ in MeCN, Table S4) into a strongly fluorescent guanidinium state in the complex (e.g., $\Phi_f = 0.42$ for **1**⊂PFOA in MeCN, Table S9).

Further experimental proof that not only protonation but also anion binding is effective can be derived from a comparison of the $\Phi_f$ data for **1**⊂PFOA in MeCN ($\Phi_f = 0.42$, Table S9), **1** in MeCN/H$_2$O 20/80 (v/v) ($\Phi_f = 0.71$, Table S4), **1** in the presence of an excess of HCl in MeCN ($\Phi_f = 0.52$; Table S9) and two model compounds in MeCN, i.e., *meso*-(*p*-(*N*,*N*-dimethylamino)phenyl)–2,6-diethyl-1,3,5,7-tetramethyl-BODIPY (**4**), **4**/HClO$_4$ and *meso*-phenyl-2,6-diethyl-1,3,5,7-tetramethyl-BODIPY (**5**, Table S8). **5**, with a non-interacting *meso*-phenyl group, can be considered as the unperturbed benchmark and has a $\Phi_f = 0.87$ in MeCN[61]. **4**, with a strong *p*-(*N*,*N*-dimethylamino)phenyl donor group in the *meso*-position, is as weakly fluorescent in MeCN as **1** ($\Phi_f \leq 0.01$) but can be switched on to $\Phi_f = 0.75$ by protonation with HClO$_4$, because the entire electronic reconfiguration through protonation occurs only at a single N atom directly attached to the phenyl group and does not involve a multi-atom group like guanidine and because the perchlorate counterion does not strongly bind to R–N$^+$(CH$_3$)$_2$H. This value is in good agreement with $\Phi_f = 0.71$ found for **1** in MeCN/H$_2$O 20/80 (v/v) (Table S4), where **1** is presumably well solvated in the aqueous mixture and does not strongly interact with an also well-solvated counterion, suggesting that $\Phi_f \approx 0.7$ is the maximal emission yield reachable for *meso*-phenyl hexaalkylated-BODIPYs carrying a protonated *p*-amino substituent at the *meso*-group. Accordingly, the distinctly lower values of **1**/HCl ($\Phi_f = 0.52$) and **1**⊂PFOA ($\Phi_f = 0.42$) in a solvent such as MeCN that allows for direct host–guest interaction strongly suggest that binding of a negatively charged species to the guanidinium unit results in lower maximal enhancement and that the position of binding is also

decisive, PFOA$^-$ involving the Y-shaped motif and the N atom attached to the phenyl group while Cl$^-$ being most likely more localized at the more positive, formally written as =N$^+$H$_2$ subunit of the guanidinium (Fig. 1a, b). The fluorescence lifetime data, $\tau_f = 5.25$ ns for **5**[61] *vs* 4.23 and 4.16 ns for **1**/HCl and **1**⊂PFOA compared to <0.01 ns for **4** and 0.03 ns for **1** (Tables S8, S9), as well as the theoretical results in Figure S5 and Table S5 also support this interpretation.

However, when taking a closer look at the titration curves in Figure S7d, which are obtained when carrying out the titration according to the first titration protocol in Section I.c.i., Supplementary Information, i.e., using spectroscopic solvents for preparation of the titrant and titrator solutions, it is apparent that the shape of the curves is unusual for a strong stoichiometric complexation. In addition, the shape with a slow initial rise also suggests that 2:1 (or 1:2) binding models do not explain the behaviour. Moreover, titrations with weaker binding carboxylates like octanoic acid (OA, see below and Figure S8) or with PFOA at much higher concentrations as used for the NMR experiments (see below and Section VIII, Supplementary Information) show again the typical behaviour of 1:1 complexes. This apparent contradiction can be understood when considering the fact that many acids undergo not only the typical deprotonation/protonation reaction in aprotic organic solvents (Eq. 1), but can also undergo homo- and heteroconjugation[62]. Homoconjugation is a specific type of dimerization (Eq. 2) that has an impact on the amount of available free (monomeric) anions for a given acid concentration and therefore also on the apparent acidity constant $K_a^{app}$ (or p$K_a^{app}$) of the acid in the non-aqueous environment (p$K_a^{app}$ because the p$K_a$ is by definition obtained in water). Heteroconjugation (Eq. 3) is the interaction of an acid with a species HR that is considered as an impurity in the system, which is in most experimental cases water that is still present in traces in the system, especially if studies are not carried out under inert conditions like is usually the case for spectroscopic measurements (in contrast to electrochemical investigations).

$$HA \rightleftharpoons A^- + H^+ \qquad (1)$$

$$HA + A^- \rightleftharpoons [AHA]^- \qquad (2)$$

$$HR + A^- \rightleftharpoons [AHR]^- \qquad (3)$$

Accordingly, depending on $K_a^{app}$, $K_{AHA^-}$ and $K_{AHR^-}$ the number of free protons provided by one molecule of a certain acid under specific conditions can vary between one and zero. In addition, the determination of these quantities is aggravated not only because they require strict control of the experimental conditions, but because they influence each other. To assess in how far the spectroscopic and photophysical complexation behaviour of **1** and PFOA might depend on the titration protocol, we screened for an alternative titration protocol, altering sequence and solvents. As the data in Fig. 3d and S9 show, we were indeed able to obtain a titration curve exhibiting a behaviour expected for strong stoichiometric complexes (with a $K_S > 10^7$ M$^{-1}$), according to the second instead of the conventional first protocol in Section I.c.i.

Time-resolved fluorescence studies carried out according to the second protocol support these findings, yet also hint into the direction of the more complex interactions (Fig. 3c–e, S9b and Section VI, Supplementary Information). Although three decay components are necessary to obtain a satisfactory global fit of the time-resolved titration data, the intermediate decay component is never exceeding a relative contribution or amplitude of $a_i^{rel} = 10\%$ and shows a similar trend vs PFOA concentration as the long lifetime component does (Fig. 3d), excluding any consecutive mechanistic relationship as would be the case for 2:1 or 1:2 stoichiometries. Accordingly, the decay associated spectra (DAS) of the intermediate and long decay

component are similar in spectral shape, and both are slightly red-shifted compared to the DAS of the short lifetime component that exemplifies free **1**. If the $DAS_i$ in Fig. 3e are spectrally fitted, the maxima for the dominant long and the minor intermediate component are shifted by $3 \pm 1$ nm (orange crosses) and $4 \pm 2$ nm (blue crosses) with respect to the short component (green crosses) attributed to **1**. Comparison of the $DAS_i$ in Fig. 3e with the species associated spectra ($SAS_i$) in Figure S9b, which are equivalent to the relative $\Phi_{f,i}$ of the three emitting species, reveals that already at 0.3 equiv. PFOA, the emission is almost exclusively represented by the strong 1:1 complex **1**⊂PFOA ($\Phi_{f,3}^{rel} = 96\%$). This pronounced difference between $DAS_i$ and $SAS_i$ is due to the largely different fluorescence lifetimes of the two major species $DAS_1$ (32 ps, free **1**) and $DAS_3$ (4.350 ns, **1**⊂PFOA), and the small contribution of the intermediate species $DAS_2$ (0.512 ns), which is tentatively attributed to an unknown complex species.

Because we were not able to find conditions for which the S-shape of the titration curves for EtOAc and PrOAc (Figure S7d) converge into the common shape as for MeCN in Fig. 3d and S9a, and because the analytically relevant experiments reported below involve **1** embedded in a polymer matrix, thus showing a microscopic binding site heterology, and always involve aqueous solutions or extracts so that homo- and heteroconjugation are negligible[63], we compared the behaviour in MeCN and the acetate solvents by analysing the data in Figure S7d with a nonlinear dose–response behaviour, which can better account for the analyte heterology observed here (for a detailed explanation, see Section VII, Supplementary Information including Figures S10–S12). If the titration curves in Figure S7d, obtained in a similar manner along the first titration protocol, are fitted accordingly, the reciprocal of the inflection point of the fitting curve is a measure for the lower limit of an apparent binding constant $K_S^{app}$, allowing to derive values of $K_S^{app} \geq 1.1 \times 10^6$ M$^{-1}$ for **1** and PFOA in MeCN as well as $K_S^{app} \geq 6.9 \times 10^5$ M$^{-1}$ and $\geq 9.4 \times 10^5$ M$^{-1}$ in EtOAc and PrOAc, respectively (see Equation S5 and Section I.f. for details). Therefore, in mechanistic terms, the unusual S-shaped binding curve in aprotic organic solvents is due to the homoconjugation of PFOA which, however, is irrelevant for the analytical assay carried out with aqueous samples. Moreover, in terms of a sensory response, the "OFF−ON" signalling behaviour of the indicator monomer is advantageous, affording a limit of detection (LOD) for PFOA of $0.43 \pm 0.03$ μM in MeCN.

As a first step to assess the selectivity of the approach, the binding of **1** to the isostructural fully hydrogenated octanoic acid (OA) was studied in MeCN (Figure S8). The interacting groups are identical, guanidine and carboxylic acid, so that the same binding mechanism was anticipated for OA and PFOA, differing only in their acidity. However, a comparison of Fig. 3a, b, d and S8 reveals that not only are significantly higher concentrations of OA required to induce spectroscopic changes, but that also the spectral shifts and maximum fluorescence enhancement are smaller, i.e., 2 nm *vs* 4 nm in absorption and emission for OA *vs* PFOA and enhancement factors of approx. 25 *vs* 40, respectively. Accordingly, the binding constant determined for **1** and OA amounts to only $K_S^{app} = 1.7 \times 10^3$ M$^{-1}$. These results indicate that **1** is less easily protonated by OA and that the hydrogen bonded partners are less strongly interacting in the complex compared to **1** and PFOA. To get further insight into the binding modes, the interaction of **1** with PFOA, HCl and OA was studied by $^1$H NMR in CD$_3$CN (Figure S13–S16). Upon addition of increasing amounts of PFOA to **1**, the NH protons, usually not detectable in the $^1$H NMR spectrum due to fast exchange processes with the solvent, appear in the spectra in the region of 7.5–11.5 ppm (Figure S16), suggesting that they are locked by formation of the ion pairing-assisted hydrogen bonded complex. In addition, the two aromatic proton signals of the *meso*-phenyl ring (H9, H10) are shifted downfield and remain clearly distinguishable in the complex, supporting tight interaction with the guanidinium NH bound to the phenyl-C4 (Figure S16). This interpretation was confirmed through $^1$H NMR titrations with HCl and OA in CD$_3$CN (Figure S16), in which low or

no NH proton signals could be observed and the signals of H9 and H10 coalesced even at higher excess of acid because of the weaker complex formation and/or the fact that for HCl, the chloride preferentially resides at the partially more positive =N$^+$H$_2$ group, which is commonly attributed to be formally positively charged (Fig. 1b). Indicator monomer **1** should thus be able to discriminate between low concentrations of organic acids already because of their acidity, converting the binding event into a bright fluorescence signal.

## Preparation and characterisation of core-shell nanoparticle probes

Although the different responses to PFOA and OA are a promising start, the fluorescent indicator alone will not be able to discriminate between strong PFCAs and other strong oxoacids that are sufficiently soluble in MeCN or acetate solvents. To impart further selectivity to the system, the indicator monomer was thus incorporated into polymeric recognition matrices that were grown from the surface of nanoparticles, a particle platform adding further functionality to and handling options for such probes in a miniaturised assay (Fig. 1c). Especially the design of the indicator monomer and the supramolecular interaction, i.e., pairing directionality through hydrogen bonds and binding strength through electrostatic attraction, should be beneficial for the generation of PFOA-tailored cavities. As outlined in Fig. 4a, silica nanoparticles (SNPs) doped with the orange-emitting fluorophore tris(2,2'-bipyridyl)ruthenium(II) chloride, Ru(bpy)$_3$Cl$_2$, **oSNPs**, were used as the carrier particles for the polymer layer, providing also the reference signal.

SNPs were chosen as core particles because they can be produced with a high monodispersity, their suspensions are usually stable with a low tendency for sedimentation and clogging and their surface can be easily functionalised with silane chemistry. In addition, water-soluble, positively charged dyes can be simply sterically entrapped in the negatively charged silica network during polycondensation if suitable methods such as reverse microemulsion techniques are chosen, contributing to the popularity of SNPs in bioapplications[64]. The present **oSNP** beads were thus synthesised by such an approach[65], yielding particles with a diameter of $57.6 \pm 0.6$ nm as determined by transmission electron microscopy (TEM, Figure S17). Moreover, steric doping with Ru(bpy)$_3$Cl$_2$ allowed the dye concentration to be adjusted to a value that provides a useful reference signal in the final customized setup in a straightforward way. Since the dye is shielded in the silica matrix its fluorescence serves as a stable internal reference signal.

For growth of the molecularly imprinted polymer (MIP) shell, the reversible addition-fragmentation chain-transfer (RAFT) polymerisation approach was followed as it offers ease of synthesis, scalability, and high uniformity, especially in connection with highly monodisperse silica particles. The polymerisation was performed on RAFT-functionalised silica nanoparticles (**raft@oSNPs**), which in turn were prepared via a two-step functionalisation of **oSNP** with (3-aminopropyl)triethoxysilane (APTES) in toluene (**a@oSNPs**) and with the RAFT agent 4-cyano-4-(phenylcarbonothioylthio)pentanoic acid (CPCTP) in tetrahydrofuran (THF) in the presence of ethyl chloroformate (EC) and triethylamine (TEA), Fig. 4a. Toluene was chosen as solvent for APTES grafting over, for instance, ethanol and acetonitrile to enhance silane condensation and reduce potential leaching of Ru(bpy)$_3$Cl$_2$.

The different particles obtained in the synthesis steps were characterised using various techniques to control the manufacturing process. N$_2$ adsorption-desorption measurements and data analysis according to the Brunauer-Emmett-Teller (BET) model revealed a surface area of $55.8 \pm 0.1$ m$^2$ g$^{-1}$ for the **oSNP** particles (Figure S18), which is important for calculation of the surface group densities during subsequent functionalisation steps. Zeta potential measurements and thermogravimetric analysis (TGA) of **oSNPs** yielded a net negative surface charge of $-22.0 \pm 0.7$ mV at pH 6 (Figure S19) for the particles, suggesting stable suspensions, and an overall mass loss of 18.3% by

a)

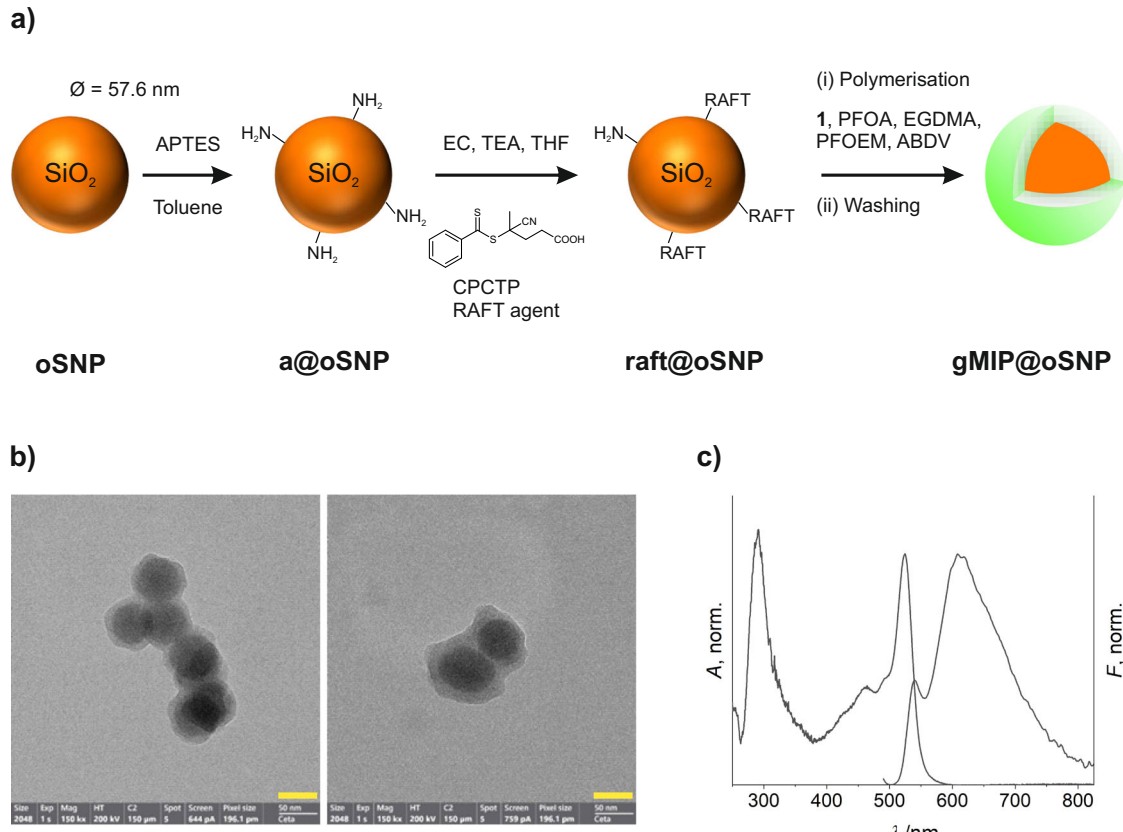

**Fig. 4 | Synthetic scheme and key characterisation of core-shell gMIP@oSNP nanoparticles for PFOA sensing. a** Scheme of **oSNP** functionalisation: i) **oSNP** are functionalised with APTES in toluene at room temperature overnight, yielding **a@oSNP**; ii) anchoring of CPCTP RAFT agent to the free amino groups of APTES on the particles' surface by acid chemistry, using EC and TEA in THF (**raft@oSNP**); and iii) growing the MIP shell from the core particles by polymerising **1**, PFOEM, EGDMA with ABDV as the radical initiator in MeCN in the presence of PFOA (**gMIP@oSNP**).

**b** TEM images of **gMIP@oSNP** with a polymer shell thickness of $17.9 \pm 0.5$ nm. Scale bars = 50 nm. **c** Corrected absorption and emission spectra of **gMIP@oSNP** (MeCN; $c_{gMIP@oSNP} = 62.5$ mg L$^{-1}$; $\lambda_{exc} = 480$ nm). APTES (3-aminopropyl)triethoxysilane, EC ethyl chloroformate, TEA triethylamine, THF tetrahydrofuran, CPCTP 4-cyano-4-(phenylcarbonothioylthio)pentanoic acid, RAFT reversible-addition-fragmentation chain-transfer, EGDMA ethylene glycol dimethacrylate, PFOEM 2-(perfluorooctyl) ethyl methacrylate, ABDV 2,2'-azobis(2,4-dimethylvaleronitrile).

TGA (Figure S20), approx. 11% of which is attributed to organic entities (alkoxy groups, dopant dye) and residual water confined in the particles; the amount of Ru(bpy)$_3$Cl$_2$ was determined to 0.3% via absorption spectroscopic measurements.

APTES functionalisation of the **oSNP** surface brought about the expected change in Zeta potential, arriving at $+21.4 \pm 0.8$ mV at pH 6 (Figure S19), and a slightly increased mass loss of 19.2% (Figure S20) for **a@oSNP**. Also as expected, further RAFT functionalisation led to a decrease of the Zeta potential ($+17.9 \pm 0.6$ mV, Figure S19) by conversion of amino into amido groups upon RAFT agent condensation, and an increase of the mass loss (22.0%, Figure S20), respectively. The sulphur content of **raft@oSNP** was determined by elemental analysis to 0.56% S, independently proving that CPCTP as the only sulphur source in this ensemble has been successfully grafted to the surface. Combining sulphur content and BET surface area, a functionalisation density of 0.95 RAFT molecules nm$^{-2}$ was calculated.

Subsequently, **raft@oSNP** were used as core particles for MIP shell growth (Fig. 4a). The polymerisation was achieved in the presence of PFOA to generate MIPs with complementary cavities, using ethylene glycol dimethacrylate (EGDMA) as a cross-linker and 2-(perfluorooctyl) ethyl methacrylate (PFOEM) as a co-monomer to further install selectivity through F−F interactions[66]. Prior to MIP synthesis, the formation and stability of the complex between indicator monomer **1** and PFOA was assessed in pre-polymerisation experiments in MeCN (Figure S21). Like the studies of **1** and PFOA in MeCN at dilute concentrations

(Fig. 3a, b), both, small red shifts in absorption and emission and a strong increase in fluorescence were observed at millimolar concentrations and in the presence of the cross-linker (EGDMA), perfluorinated co-monomer (PFOEM) and carrier particles (**raft@oSNP**), proving that the complex is also formed under conditions relevant for MIP preparation. MIP shell formation was thus initiated using 2,2'-azobis(2,4-dimethylvaleronitrile) (ABDV) as a radical initiator in MeCN, yielding the final particle probes **gMIP@oSNP** after 19 h of polymerisation. Details on MIP shell synthesis are given in Section IX, Supplementary Information, including Table S10.

**gMIP@oSNP** were characterised by TEM and optical spectroscopy, showing core-shell particles with a shell thickness of $17.9 \pm 0.5$ nm, arriving at a total average diameter of approx. 93 nm for the hybrid particle probes (Fig. 4b). Furthermore, **gMIP@oSNP** showed the typical absorption and fluorescence features of both fluorophores, Ru(bpy)$_3$Cl$_2$ and BODIPY, upon excitation at 480 nm (Fig. 4c). The band maxima of the dye in the particle core were at around 460 nm and 615 nm, respectively, and those of the dye in the MIP shell at around 525 nm and 540 nm, consistent with the large and small Stokes shifts of the fluorophores. To assess whether our approach of combining two dyes with largely different Stokes shifts to avoid potentially interfering energy transfer processes at the core-shell interface has paid off, theoretical and experimental investigations were performed, which are discussed in detail in the context of Figures S22, S23 and Table S11, Supplementary Information. These results

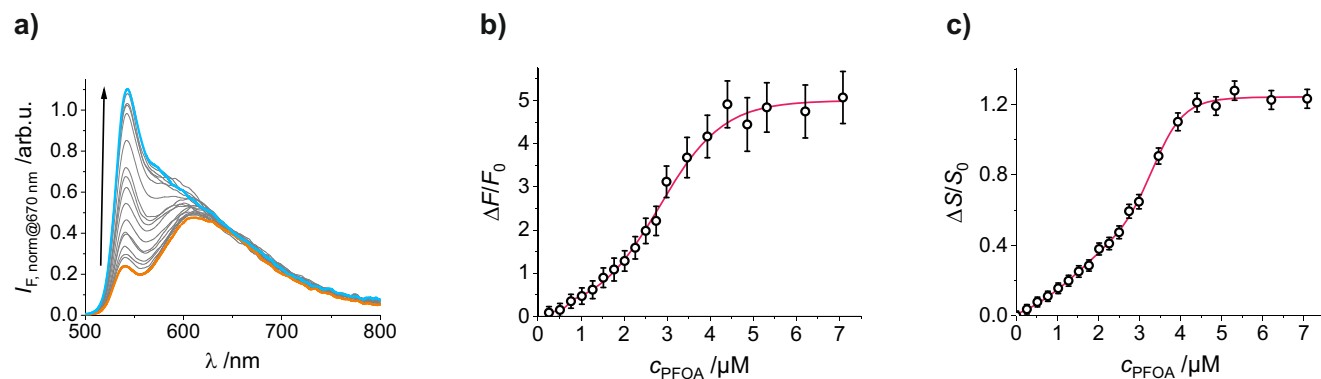

**Fig. 5 | Fluorescence response and signal quantification of gMIP@oSNP upon PFOA binding.** Fluorescence titration of **gMIP@oSNP** with PFOA: **a** titration spectra, normalized at 670 nm; **b** change in fluorescence of **gMIP** shell, determined after deconvolution and integration of the spectra in a), see Figure S25 and text; **c** corresponding signal change $\Delta S/S_0$, see text (MeCN; $c_{gMIP@oSNP} = 62.5\,\text{mg L}^{-1}$; $\lambda_{exc} = 480\,\text{nm}$). For (**b**, **c**), data are presented as measurement uncertainties, see Eq. S1b ($n_r = 3$ independent experiments).

showed that energy transfer between the dyes in the two compartments is highly unlikely and could not be found, contributing to the simplicity of the approach and the straightforward use of the emission from the core as an internal reference.

### Sensing performance of gMIP@oSNP particles

After successful MIP polymerisation, the response behaviour of **gMIP@oSNP** was investigated in MeCN. Upon addition of PFOA into a suspension of **gMIP@oSNP** in MeCN under stirring, the emission intensity change is virtually instantaneous (on a time scale of seconds) with the signal remaining stable for at least several minutes (signal decrease of approx. 4.5% after 10 min, Figure S24). For unequivocal quantification and internal referencing of a titration with for instance PFOA, the bands of **gMIP@oSNP** were first normalized at 670 nm, in the **oSNP** band, then deconvoluted and finally were the integrals of the **gMIP** spectra calculated (Figure S25). Plotting the spectra or integrated intensities as a function of increasing concentrations of PFOA yields a 5-fold increase of the typical BODIPY emission signal, and a red shift of the band maximum from 540 to 544 nm (Fig. 5a). The lower maximum fluorescence enhancement compared to neat **1** is most likely due to the constraints experienced when the fluorophore is incorporated into the polymer matrix, which reduces molecular motions and other paths of non-radiative deactivation. Interestingly, the obtained titration curve showed a bimodal dose–response behaviour (Fig. 5, S25), which is presumably due to indicator molecules being incorporated in the polymer network as well as closer to and at the surface of the polymer shell thus showing a bimodal distribution of microscopic binding constants and resulting in a more complex response than for neat **1**, see Section I.g. in Supplementary Information for further details.

To obtain further insight into the selectivity and the role of acidity, **gMIP@oSNP** were titrated with octanoic acid (OA) and the sodium salt of PFOA (PFOA-Na) (Figure S26), and the obtained results were compared with those of PFOA using the so-called discrimination factor (*DF*), Eq. 4

$$DF = \frac{\Delta F/F_0(template)}{\Delta F/F_0(other)} \qquad (4)$$

where $\Delta F/F_0$(template) and $\Delta F/F_0$(other) are the respective changes in fluorescence intensity of the **gMIP** shell upon binding to the imprinted analyte ( = template) as well as the other analytes at a specific concentration, respectively; a definition of the quantities is given in the *Methods* section below.

Here, the *DF* for 5 μM of analyte was determined to 300 and 13 for PFOA against OA and PFOA-Na, respectively, highlighting on one hand the selectivity of the MIP shell and on the other hand the importance of the presence of a strong acid to convert the guanidine form of **1** into its guanidinium form; if the sodium salt is used, the response is significantly smaller. The *DF* determined for the saturation concentration of OA is 2.5, but such millimolar concentrations of medium- or long-chain aliphatic carboxylic acids are not relevant in the case of real samples[67].

Taking advantage of the doped core as internal reference, the fluctuations of particle concentration over time or between repeat experiments could be accounted for as reflected by a drastic decrease of the calculated total relative uncertainties of the measurements (Figure S27)[68]. For the absolute signal intensity corresponding only to the integrated shell emission of the BODIPY ($F_{BDP}$: 530–624 nm), a total relative error of 30% was found, stemming almost exclusively from the repetition errors. Differential analysis ($\Delta F/F_0$) allowed for reduction of the error to 10%, accounting for offset differences between several titrations and the biggest contributor, the repetition error. Finally, the inclusion of the integrated core emission intensity of doped $Ru(bpy)_3Cl_2$ ($F_{RBP}$: 625–750 nm) as a reference led to a further reduction of the analysis error, down to 5% when only using the ratiometric shell/core intensity-based correction (*S*, Eqs. 5), and 2.8% when using the combination of the differential and ratiometric correction ($\Delta S/S_0$, Eq. 6). The LOD of the sensory particles was determined to 0.47 μM for PFOA in MeCN.

$$S = \frac{F_{BDP}}{F_{RBP}} = \frac{\int_{530}^{624} I_{F,BDP}(\lambda)d\lambda}{\int_{625}^{750} I_{F,RBP}(\lambda)d\lambda} \qquad (5)$$

$$\Delta S = S_x - S_0 \qquad (6)$$

Here, the integrals refer to the spectral ranges of BODIPY and $Ru(bpy)_3Cl_2$, see above, $S_x$ is the referenced signal obtained for a certain concentration $x$ of an analyte and $S_0$ is the respective signal in the absence of an analyte, see also *Signal Processing* in *Methods* below.

Because the addition of an aqueous sample to an MeCN suspension of **gMIP@oSNP** is not advantageous in maintaining a small measurement uncertainty, see discussion in the context of Figure S4b above, the ability of **gMIP@oSNP** to recognise PFOA was next studied in EtOAc as a water-immiscible non-halogenated organic solvent that is well compatible with a phase transfer extraction of the analyte from an aqueous sample solution. Moreover, to be able to detect PFOA as well

a)

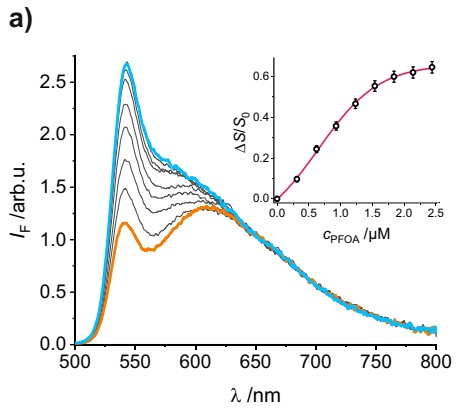

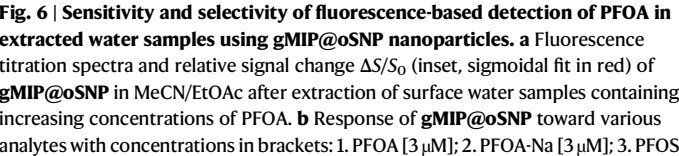

b)

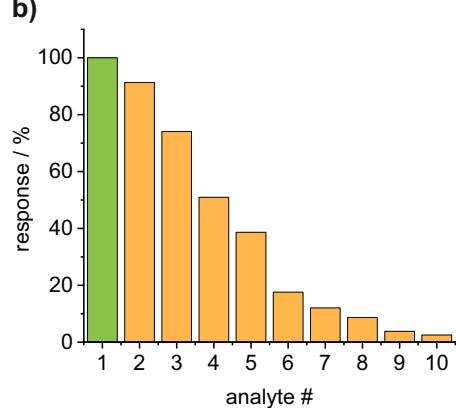

**Fig. 6 | Sensitivity and selectivity of fluorescence-based detection of PFOA in extracted water samples using gMIP@oSNP nanoparticles. a** Fluorescence titration spectra and relative signal change $\Delta S/S_0$ (inset, sigmoidal fit in red) of **gMIP@oSNP** in MeCN/EtOAc after extraction of surface water samples containing increasing concentrations of PFOA. **b** Response of **gMIP@oSNP** toward various analytes with concentrations in brackets: 1. PFOA [3 μM]; 2. PFOA-Na [3 μM]; 3. PFOS

[3 μM]; 4. SDBS [3 μM]; 5. SDS [3 μM]; 6. AcOH [200 μM]; 7. OA [100 μM]; 8. AMOX [3 μM]; 9. ENOX [3 μM]; 10. OA [3 μM] ($c_{\text{gMIP@oSNP}}$ = 62.5 mg L$^{-1}$; $\lambda_{\text{exc}}$ = 480 nm). For the inset in (**a**), data are presented as measurement uncertainties, see Eq. S1b ($n_r$ = 3 independent experiments). PFOS perfluorooctanesulphonic acid, SDBS sodium dodecylbenzenesulphonate, SDS sodium dodecyl sulphate, AcOH acetic acid, AMOX amoxicillin, ENOX enoxacin.

as its anionic carboxylate form, which is prevalent in aqueous solution, the water samples were acidified to pH 2 with HCl before the extraction step. In monophasic EtOAc, PFOA concentrations down to 0.29 μM could be detected with relative errors of 10% (Figure S28). However, as for indicator monomer **1**, for which the maximum signal enhancement was 18-fold *vs* 39-fold in EtOAc *vs* MeCN, see above, a distinctly lower maximum signal enhancement was also found for **gMIP@oSNP** in PrOAc or EtOAc *vs* MeCN, i.e., $\Delta S/S_0$ = 0.26 or 0.35 *vs* 1.24. In addition, the value for EtOAc was still slightly lower (< 0.20) in water-saturated EtOAc after liquid-liquid extraction (Figure S28 and Table S12). Remarkably, when surface water at pH = 2 (HCl) was spiked with 0.33 μM of PFOA, 96% of the analyte were recovered and detected in the organic phase. Nonetheless, we wished to increase the sensitivity of the assay and envisioned that the addition of MeCN to the EtOAc phase while retaining immiscibility with water should be a viable option. We thus conceived a biphasic ternary MeCN/EtOAc/H$_2$O solvent system and studied the phase separation behaviour of an organic phase (O) consisting of *x* % MeCN and (1−*x*) % EtOAc and an aqueous phase (W) in a ratio of O/W = 1/1 (v/v). For MeCN proportions higher than 50 vol% in the organic phase, an undesired partial mixing of the phases occurred, restricting the working range to *x* < 50% (Figure S29a). When taking a closer look at the behaviour of both **1** and **gMIP@oSNP**, for **1** alone, a decrease of the $\Delta F/F_0$ upon addition of 10 μM PFOA could be observed upon increase of MeCN proportion, presumably due to an uptake of a larger amount of H$_2$O (pH 2, HCl) into the organic phase. In contrast, the **gMIP@oSNP** signal followed an opposite trend, with a linear increase of $\Delta S/S_0$ with the MeCN proportion upon addition of 10 μM PFOA (Figure S29b). This linear increase with increasing MeCN proportion suggests that despite more water being taken up into the organic phase, the MIP layer might be preferentially solvated by MeCN and/or EtOAc, MeCN having been used as the porogen in the polymerisation step.

Although the results of the ternary system were encouraging, for practical purposes, we separated the extraction from the detection step. PFOA was thus first extracted from an acidified (pH = 2) aqueous sample into EtOAc, minimizing the uptake of water in the organic phase and maximizing the extraction efficiency (Figures S29), before the EtOAc phase containing the extracted analyte was mixed with a suspension of **gMIP@oSNP** in MeCN in a 1/1 (v/v) ratio. Following such a protocol, an LOD of 0.18 μM and a maximum signal change $\Delta S/S_0$ of 0.63 were obtained for PFOA in Milli-Q water (pH = 2) as the sample, respectively (Figure S30). Moreover, when changing the matrix from

Milli-Q water to surface water collected from the Teltow Canal (52°25′32.5″N, 13°32′15.9″E, Berlin, Germany) and used without further treatment other than filtration and acidification at pH = 2, negligible matrix effects were observed (Fig. 6a), yielding an LOD of 0.11 μM and a maximum signal change $\Delta S/S_0$ of 0.64 for PFOA, respectively. Since PFOA exists in its anionic form in aqueous samples, next the response toward an aqueous sample in which PFOA-Na was dissolved was studied, revealing good agreement with PFOA in its neat form (Fig. 6b). The cross-sensitivity of the **gMIP@oSNP** particle probes against other species and several interferents was subsequently assessed along the same assay protocol. As can be seen in Fig. 6b, perfluorooctanesulphonic acid (PFOS) induced almost 75% of the response observed for PFOA, but SDBS and SDS induced less than 50% response. Higher concentrations of OA and AcOH, the latter to simulate short-chain acids that can be found in rain or surface waters[69,70], showed only low interferences of up to approx. 15%, concentrations of 100 μM of medium- or long-chain carboxylic acids (entry #7 for OA in Fig. 6b) already exceeding the background concentration of natural waters by >10-fold[67].

To better understand the response pattern shown in Fig. 6b, it is helpful to consider the single steps contributing to the response and how they are determined by molecular properties and structural features of the analytes and competitors. For this purpose, Table S13 collects all the relevant data and Figure S31 the chemical structures of the respective species.

The first step is the extraction step. Since no data are available for the binary and ternary biphasic solvent systems used here, this step is perhaps best approximated by the octanol-water partition coefficient, which is usually expressed as log$K_{\text{OW}}$. Except for PFOS, for which most regulatory agencies state that a log$K_{\text{OW}}$ cannot be reliably determined and for which we took one of the few calculated values reported (which in itself are quite divers, e.g., −1.08[71] or 4.49[72]; in analogy to SDS, we opted for −1.08 here), the data suggest that OA, PFOA and SDBS might be better extracted than PFOS, SDS and the other organic acids. However, it should be noted that log$K_{\text{OW}}$ can dramatically depend on the pH of the aqueous phase, increasing as the amount of anion decreases thus contributing to the uncertainty of the data available.

The second step of importance for generating a strong analyte-induced signal is protonation of the MIP-bound indicator monomer, which is best approximated by the p$K_a$ as primary driver for ionic interaction. This is also only an approximation because protonation of **1** happens in the MIP matrix filled with the organic phase and the

apparent p$K_a$ values in organic solvents are significantly different from those in water, although the general trends are usually preserved. Here, clearly, PFOS, for which again only a calculated value is available, and SDBS are the most acidic, followed by PFOA and SDS.

The third factor that is influencing the fluorescence response is the possibility of the binding units of **1** and the oxoanion being able to optimally arrange for tightest possible hydrogen bonding. This parameter is best expressed by the tilt angle between the Y-shaped H-bond binding motif(s) of the acid head group of a guest and the rest of the molecule that would have to fit into the PFOA-imprinted cavity. Thus, it is the first parameter that is determined by the MIP as the chemical recognition element and the strategy of directional imprinting of a complex through arrangement of the host-guest ion pair via two hydrogen bonds. The data in entries #3 in Table S13 show that the planar carboxylate head groups in PFOA and OA are well aligned, showing a tilt angle of 31°, whereas the tetrahedral sulph(on)ate head groups in PFOS, SDBS and SDS and the carboxylate head groups in the three other oxoanions are less optimally positioned with respect to the fit-in-cavity directing bulk shape of the molecule.

The fourth, sixth and seventh factor are related to the rest of the molecule, i.e., the entire molecule except for the acid head group, either determining the stabilisation through F−F interactions with the fluorinated co-monomer (entries #6) or the fit of the guest into the imprinted cavity, i.e., the difference in length and shape of the guest compared with PFOA as the imprinted template. Favourable F−F stabilisation is seen for PFOA and PFOS, yet SDBS and SDS should also experience a sizable stabilisation because their longer chains offer more points for interaction. However, this interaction is counter-balanced for SDBS and SDS by their distinctly longer size, yet with a (roughly) similar shape as straight (SDS) or kinked (SDBS) cylinders. OA shows the same length and shape features as PFOA yet lacks the strong F−F stabilisation. All the other oxoanions lack stabilisation through F−F interactions as well as through size and shape. Of course, these parameters all depend critically on how uniform and homogeneous the cavities in the MIP have been formed.

In view of the rather approximative character of the parameters, a comparison of the number (and brightness) of the green entries in Table S13 with the response in Fig. 6b should best allow to derive trends and interpret these findings. If we start from the least responding analytes, ENOX, AMOX and AcOH, it is evident that they are inferior in terms of the three key parameters log$K_{OW}$, p$K_a$ and the tilt angle. While in case of AMOX and ENOX the molecular size and/or shape is also rather unfavourable to bind to a PFOA MIP, AcOH is too small to experience sufficient stabilisation in the cavities.

A comparison of OA (four greenish entries) and SDS (three greenish entries) shows that the difference in response between entry #5 (38% signal intensity for SDS compared with PFOA) and #10 (2% signal intensity for OA compared with PFOA) in Fig. 5b is not due to log$K_{OW}$, where OA shows a much more favourable behaviour than SDS (five orders of magnitude), but rather suggests that for the low concentrations used in our studies while acidifying the aqueous phase to pH 2, the partition coefficient can be largely disregarded and even a more water-soluble species such as SDS is presumably quantitatively extracted into the organic phase. For instance, ethyl acetate extraction at pH 1.5 has also been used successfully by others for oxoanion determination[73]. The main factor in favour of SDS should be the more than three orders of magnitude lower p$K_a$.

Based on the main design considerations, the largely different response between PFOA and OA is thus mostly due to p$K_a$ (almost four orders of magnitude difference), i.e., to the protonation of **1** and subsequent hydrogen bonded complex formation, with a certain contribution from F−F stabilisation in the MIP.

PFOA and PFOS are rather similar in size, shape and F−F stabilisation, yet differ in tilt angle and p$K_a$. While the almost four orders of magnitude lower p$K_a$ of PFOS should dramatically favour a response

from this species over PFOA, the significantly worse interaction possibilities of the sulphonate head group with the guanidinium moiety that has been positioned in the MIP via directional PFOA imprinting seems to be able to counterbalance this force (the PFOS response is 75% of the PFOA response), stressing the potential that MIP-type receptor cavities containing an indicator monomer can have on fluorescence signal generation.

The other two non-fluorinated long-chain surfactants should experience a certain weaker stabilisation through their shape, size and the lack of F−F interactions, while also possessing a less strongly acidic sulph(on)ate head group than PFOS, yet while still being stronger or almost equally strong acids than PFOA, which overall leads to weaker responses of 51% (SDBS) and 38% (SDS) compared with the PFOA response. Thus, the sequence PFOS (75%) > SDBS (51%) > SDS (38%) seems to be mainly governed by p$K_a$ with contributions of the structural and F−F interactions forces imposed by the MIP.

## Opto-microfluidic platform for PFOA detection from water samples

The integration of fluorescent MIP particles in miniaturised microfluidic devices is attractive as it enables portable, fast, and automated on-site detection. Here, a modular opto-microfluidic setup was constructed from a commercially available mixing chip and PFA (perfluoroalkoxyalkane) tubing (Figure S32). The EtOAc extraction solution containing the analyte (e.g., PFOA) and the suspension of the **gMIP@oSNP** particle probes in MeCN were flown in a 1/1 (v/v) ratio through an inert Topas pearl chain mixer for mixing by advection. The mixture was then guided into a highly transparent PFA tube, allowing for the direct collection of the fluorescence signals. The flow rate was set to 11 μL min$^{-1}$ for both solutions, ensuring mixing in the passive mixer for 7 min (chip volume of 153 μL) and a total transition time from syringe to detection point of 10 min (tubing volume of approx. 65 μL, see Section XII, Supplementary Information, including Figures S32, S33 for more details).

In the fluorescence detection module, the PFA tube passed through an opto-mechanical cube with two oppositely mounted laser diodes (LD) as excitation sources for green and orange emission of **gMIP@oSNP**, i.e., an LD at 450 nm for the orange **oSNP** reference signal and an LD at 520 nm for the green emission of the BODIPY indicator in the MIP shell. A collection channel was mounted perpendicularly to the excitation channels with a lens for coupling of the emission signals into an optical fibre bundle connected to a USB spectrometer. An acquisition time of 2 min was sufficient to obtain a stable fluorescence signal with acceptable uncertainty. Thanks to the spectral resolution of the spectrometer, the signals could be easily separated and $\Delta S/S_0$ calculated, allowing to eliminate any fluctuations not related to the analyte concentration like a change in the particle concentration due to sedimentation. As in the cuvette experiments, the internal reference allowed for a strong reduction of measurement uncertainty because the repetition errors could be reduced from 20% to 10%. Upon increasing concentrations of PFOA in surface water samples, extracted from 8 mL into 4 mL EtOAc through shaking in a vial for 2 min after acidification to pH 2 with HCl (1 M) and injected into the microfluidic system, the gradual enhancement of the fluorescence signal from **gMIP@oSNP** particles could be registered (Fig. 7). With this dedicated setup and a corresponding calibration curve, an LOD of 1.2 μM was obtained for PFOA, with no apparent interferences from matrix substances contained in the surface water samples. In addition, memory effects due to the tubing or chip were also not observed (Figure S34).

## Discussion

A guanidine-functionalised BODIPY indicator monomer was synthesised for the detection of PFOA from water. Host-guest interaction occurred via concerted protonation-mediated ion pairing-assisted

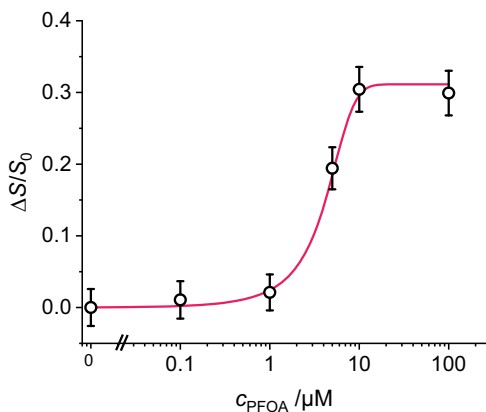

**Fig. 7 | Quantitative fluorescence response of gMIP@oSNP to PFOA in microfluidic assay of extracted surface water samples.** Relative signal change of **gMIP@oSNP** particles as a function of increasing concentrations of PFOA spiked into a surface water sample, after acidification of the aqueous sample and injection of the EtOAc extract into the microfluidic system together with a MeCN suspension of the particle probes ($c_{gMIP@oSNP}$ = 62.5 mg L$^{-1}$; $\lambda_{exc}$ = 450, 520 nm). Data are presented as measurement uncertainties, see Eq. S1b ($n_r$ = 3 independent experiments).

hydrogen bonding, combining the strength of ionic interactions with the directionality of hydrogen bonding. This binding mode significantly enhanced the fluorescence, enabling "OFF−ON" detection in polar organic solvents, and imparted the system with a first means of selectivity, i.e., indicating strong *vs* weak organic oxoacids. To endow the approach with a second means of selectivity, the pre-arranged complex between guanidine BODIPY indicator monomer and PFOA analyte was integrated into a MIP layer which was grafted from the surface of tris(bipyridine)ruthenium(II) chloride-doped silica nanoparticles, the latter constituting an internal reference signal that improves assay robustness and significantly reduces measurement uncertainty. Nanoparticles were chosen as the format here because they can be handled in a straightforward manner in microfluidic systems, avoiding fast sedimentation or clogging. Selective detection of PFOA from environmental water samples was successful in a straightforward extract-&-detect assay after combination of the MIPs with a liquid-liquid extraction step and acidification of the aqueous sample, providing the third means of selectivity. This approach yielded a favourable LOD of 0.18 μM and 0.11 μM for PFOA from Milli-Q water and surface water samples, respectively.

The study revealed valuable insight into the various aspects that govern the performance of a fluorescent molecularly imprinted polymer-based sensing system for PFCAs. It showed that although forces such as the protonation of a host (indicator monomer **1**) by a guest (oxoacid) and the subsequent ionic interactions are usually much more dominant than all the other non-covalent forces, especially the optimal positioning of host and guest in an imprinted network is able to counterbalance a p$K_a$ difference of four orders of magnitude. The latter was only possible through the installation of the second supramolecular binding process, hydrogen bonding that, in concert with electrostatic attraction, allows to favour binding of an organic carboxylate over a sulph(on)ate.

The extract-&-detect assay with MIPs was integrated into a modular opto-microfluidic setup creating a compact, user-friendly, and cost-effective detection system. This system allows for rapid ( < 15 min) and sensitive on-site detection of PFOA in the lower micromolar range. Compared to existing technologies, our sensing system provides several distinct advantages such as direct indication via host-guest interaction, high selectivity to strong hydrophobic organic acids, effective matrix suppression, minimum sample treatment, fast analysis time and

low reagent consumption, including organic solvents (Table S14). These features address the critical need for on-site and rapid detection methods, especially in the context of increasing environmental concerns regarding PFAS contamination and is already now suited for use in contaminated suspect sites. The modular design also supports integration with further enrichment steps, for instance, together with the extraction step directly in the device, to realise even lower detection limits and enhance the discrimination against organic sulph(on) ates, providing a significant advancement in the development of environmental monitoring and analysis tools for PFOA. Future research will explore the utility of this system for a broader range of PFAS compounds, aiming to develop class-based approaches and strategies to increase selectivity and sensitivity. The use of photon-counting USB photomultiplier (PMT) modules with miniaturised lock-in amplifiers could further increase sensitivity.

## Methods

Details on chemicals and materials, instrumentation methods, titration protocols and analytical figures can be found in Section I, Supplementary Information.

### Synthesis of 2-(3-(4-(2,6-diethyl-1,3,5,7-tetramethyl-4,4-difluoro-4-bora-3a,4a-diaza-(s)-indacenyl-phenyl)-guanidino)-ethyl methacrylate (1)

To a solution of 2-(3-(4-(2,6-diethyl-1,3,5,7-tetramethyl-4,4-difluoro-4-bora-3a,4a-diaza-(s)-indacenyl)phenyl)thioureido)-ethyl methacrylate **2** (200 mg, 0.35 mmol, see ref. 57 and Section II, Supplementary Information for synthetic details) and N,N′-dicyclohexylcarbodiimide (740 mg, 3.6 mmol) in anhydrous CHCl$_3$ (30 mL), ammonia solution (7 N in methanol, 1.0 mL) was added dropwise. The reaction mixture was stirred at room temperature for approximately 5 d until the starting material was totally consumed (monitored by TLC). The solvent was removed under vacuum, and the residue was purified by column chromatography on silica gel (EtOAc/CH$_2$Cl$_2$, 1:1−1:0), followed by recrystallisation in CH$_2$Cl$_2$/hexane to give the desired indicator monomer **1** as a red powder (140 mg, 72%).$^1$H NMR (400 MHz, CDCl$_3$): δ (ppm) = 7.16 (d, $J$ = 8.4 Hz, 2H), 7.02 (d, J = 8.3 Hz, 2H), 6.15 (s, 1H), 5.61 (s, 1H), 4.36 (t, $J$ = 5.4 Hz, 2H), 4.06 (s, 3H), 3.63 (t, $J$ = 5.4 Hz, 2H), 2.52 (s, 6H), 2.30 (q, $J$ = 7.5 Hz, 4H), 1.97 (s, 3H), 1.37 (s, 6H), 0.98 (t, $J$ = 7.5 Hz, 6H). $^{13}$C NMR (101 MHz, CDCl$_3$): δ (ppm) = 167.76, 153.51, 150.66, 150.42, 140.98, 138.52, 136.20, 132.75, 131.28, 129.50, 129.46, 126.23, 126.20, 123.99, 63.99, 41.08, 18.51, 18.45, 17.22, 14.76, 12.60, 12.01, 11.95. HRMS(ESI + ): $m/z$ calculated for C$_{30}$H$_{39}$BF$_2$N$_5$O$_2$ [M + H]$^+$: 550.3165, found: 550.3124.

### Preparation and functionalisation of orange tris(bipyridine) ruthenium(II) chloride-doped silica nanoparticles (oSNPs)

The synthesis of the orange-fluorescent Ru(bpy)$_3$Cl$_2$-doped silica nanoparticles **oSNPs** was scaled up from a reported reverse microemulsion method[65,74]. Concisely, a water-in-oil microemulsion was prepared by mixing Triton X-100 (17.7 mL, 18.8 g), n-hexanol (18 mL), cyclohexane (75 mL), tris(bipyridine)ruthenium(II) chloride in water (4.8 mL; 17 mM) and TEOS (1 mL) in a hybridisation glass tube. After the addition of TEOS the mixture was vigorously stirred. Then, after mixing for further 20 min in the hybridisation oven, NH$_3$ (32% aq. solution, 600 μL) was added to initiate the polymerisation. The reaction was allowed to continue for 24 h at room temperature in the hybridisation oven. After completion of the reaction, the **oSNPs** were isolated from the microemulsion using acetone and centrifugation (9961 × g); and washed by centrifugation (9961 × g) and redispersion with 2 × 45 mL ethanol (96%), 2 × 45 mL water and 2 × 45 mL ethanol (96%) to remove any surfactant molecules and non-entrapped dye molecules until a clear supernatant was obtained. The **oSNPs** were finally dried overnight in a vacuum at room temperature to obtain an orange solid residue. Ultrasonication was used during the washing steps to

remove any physically adsorbed fluorophores from the particle surfaces.

Functionalisation of **oSNPs** with APTES and, subsequently, the RAFT agent CPCTP was carried out as we have reported earlier, yielding the RAFT agent-functionalised particles **raft@oSNP**[44,45]. Further details on synthesis and characterisation are given in Sections IX and X, Supplementary Information.

### Preparation of orange silica nanoparticle core/green molecularly imprinted polymer shell particle probes (gMIP@oSNP)

For coating a thin MIP layer onto the orange-fluorescent $Ru(bpy)_3Cl_2$-doped silica nanoparticles **oSNPs**, a complex between the green indicator monomer **1** (1.51 mg, 2.8 µmol) and the target analyte PFOA as the template (1.14 mg, 2.8 µmol) was formed in 3.08 mL MeCN in a 4 mL brown vial (see Table S10). Afterwards, ethylene glycol dimethacrylate (EGDMA, 25.8 µL, 136 µmol) and 2-(perfluorooctyl)ethyl methacrylate (PFOEM, 8.1 µL, 24.2 µmol) were added to the complex mixture after having been filtered through inhibitor removers to remove the hydroquinone monomethyl ether radical inhibitor. In a subsequent step, the solution was added to 20 mg of reversible addition-fragmentation chain-transfer (RAFT) agent-coated **raft@oSNP** particles (see Section IX, Supplementary Information) and sonicated until the particles were suspended and no agglomerate layer could be observed at the bottom of the brown vial. Afterwards, 50 µL of MeCN containing ABDV (0.6 mg, 3.3 µmol) were pipetted into the mixture while purging with Argon and cooling ($-18\,°C$). The reaction vial was then wrapped with Parafilm and aluminium foil. Finally, the mixture was left to react for 18 h at 55 °C under magnetic stirring (750 rpm) and an inert atmosphere and, additionally, 1 h at 70 °C.

After the reaction was complete, the **gMIP@oSNP** particles were isolated using hexane (10 mL) and centrifugation at 13312 ×g; and washed by centrifugation and redispersion with 4× acetonitrile (10 mL) to remove the imprinted template and any unreacted complex, co-monomer, cross-linker, or initiator molecules. The **gMIP@oSNP** particles were finally dried overnight in a vacuum at room temperature to obtain an orange solid residue.

### Signal processing

Unless otherwise stated, each measurement was realised in triplicate (detailed titration protocols in Section I, Supplementary Information). For titration with **1**, the spectral band parameters ($I_{F,max}$, $\lambda_{max}$, and area $F$) were directly extracted from the raw fluorescence emission spectra after blank correction without any further mathematical treatment. The fluorescence changes were derived from the integrated areas of the emission spectra between a certain start and end wavelength according to Eq. 7

$$F = \int_{start}^{end} I_F(\lambda)d\lambda \qquad (7)$$

and usually plotted as reduced fluorescence intensity changes $\Delta F/F_0$ as given by Eq. 8

$$\Delta F/F_0 = (F_x - F_0)/F_0 \qquad (8)$$

where $F_0$ is the integrated fluorescence intensity in the absence of the analyte and $F_x$ is the fluorescence intensity for each titration step with the analyte at concentration $x$.

The ratiometric fluorescence signals $S$ obtained and analysed for the dual fluorescent particle probes follows the same rationale, i.e., $S$ is the ratio of the integrated fluorescence areas $F$ of BODIPY and $Ru(bpy)_3Cl_2$ and $\Delta S$ and $\Delta S/S_0$ are the respective subtracted and reduced intensities in analogy to Eq. 8.

The relative uncertainties of the measurements were calculated for each concentration of analyte taking in account the errors $u_i$

coming from the preparation of the solutions, instrumental errors and the repetition of the experiments according to Eq. 9, see Section I.d., Supplementary Information for details[68].

$$u_{tot} = \sum \sqrt{(n_i \times u_i)^2} \qquad (9)$$

The limit of detection (LOD) describing the smallest concentration of analyte that can be reliably detected was determined by computing first the limit of blank (LOB) and secondly the LOD from the LOB and the standard deviation of the sample with the lowest concentration (Eq. 10), see Section I.e., Supplementary Information, according to Armbruster and Pry[75].

$$LOD = LOB + 1.645(SD_{lowest\ conc\ sample}) \qquad (10)$$

While the LOD is the relevant quantity describing the detection performance of the last measurement step, the total relative measurement uncertainty $u_{tot}$ is a more realistic quantity for describing the overall performance of an assay. When developing analytical assays with custom materials and devices, we recommend deriving both quantities as they give a much better overview of the contributions of the individual assay steps to the uncertainty budget of the method and thus facilitate its optimization.

## Data availability

The authors declare that all the data supporting the findings of this manuscript are available within the manuscript, supplementary information, source data file and from the corresponding author upon request. Source data are provided with this paper.

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

## Acknowledgements

The authors thank S. Rother (BAM) for synthetic and characterisation support, K. Keil (BAM) for molar absorption coefficient determination, M. Grüneberg (BAM) for TGA, A. Zimathies (BAM) for BET surface area, J. Lisec (BAM) for HRMS, K. Meyer (BAM) for NMR, A. Meckelburg (BAM) for EA, J. Riedel and M. Koch (BAM) for LC-MS/MS measurements. This work has received funding from the European Union's Horizon 2020 research and innovation programme under the Marie Skłodowska-Curie Action grant agreement (GA) No 721297 (GlycoImaging, V.P.-P.) and the GA No 848098 (REVERT, V.P.-P. and V.V.). We thank the China Scholarship Council for a fellowship to Y.S. (No. 201908330307).

## Author contributions

CRediT: Y.S.: conceptualisation, formal analysis, funding acquisition, investigation, methodology, visualisation, writing – original draft. V.P.-P.: conceptualisation, formal analysis, investigation, methodology, visuali-sation, writing – original draft. V.V.: methodology, supervision, writing – review & editing. J.B.: conceptualisation, formal analysis, investigation, methodology, supervision, visualisation, writing – review & editing. K.G.: conceptualisation, investigation, supervision, writing – review & editing. K.R.: conceptualisation, formal analysis, funding acquisition, metho-dology, resources, supervision, visualisation, writing – review & editing.

## Funding

## Competing interests

One of the authors, V.V., is an editor on the staff of Nature Commu-nications, but was not in any way involved in the journal review process. The other authors declare no competing interests.
