## [Transparent Peer Review file · Nature Communications]

Ratiometric detection of perfluoroalkyl carboxylic acids using dual fluorescent nanoparticles and a miniaturised microfluidic platform

Corresponding Author: Dr Knut Rurack

Version 0:

Reviewer comments:

Reviewer #1

(Remarks to the Author)

To my opinion, Rurack et al. has established a very interesting method to detect perfluoroalkyl carboxylic acids, a family of emerging pollutants whose concern is growing rapidly and, for this reason, there is a huge demand, even rightly coming from legislators, to check their concentrations in water also using simple approaches. Reading this article, I have been impressed by the high level of detail. The thorough explanations, comprehensive analysis, and meticulous attention to every aspect have yielded a high level article. The authors made a very extensive and accurate work reaching promising results, and for all these reasons I suggest its publication although not in the present form. I have in fact a major concern, that is related to the shape of the titration profiles such the one reported in the inset of figure 2b. A pure 1:1 stoichiometry, even with high association constants, to my knowledge, should not lead to an upward curvature in the first part of the titration, so that, if I am not wrong, this behaviour could hide another process that could be worth to investigate.

As a very minor point, I would also suggest not to stress — in the introduction — that previous approaches could have, as possible drawback, the use of “significant amounts of organic solvents” or, better, to make a lower use of organic solvents among the possible improvements of their approach mentioned in the last sentence of the discussion.

Reviewer #2

(Remarks to the Author)

Rurack and coauthors reported synthesis of a novel BODIPY indicator monomer functionalized with green fluorescent guanidine, combined with molecularly imprinted polymers (MIPs) for the detection of PFOA in water samples via liquid-liquid extraction. While the mechanism behind the fluorescence enhancement is discussed, there are several limitations in this study. The research findings do not fully support the claims made and lack significant innovation, leading to the recommendation against publication in nature communications.

1. The discussion of the mechanism of fluorescence signal change following the interaction between the analyte and the receptor lacks supporting evidence from characterization methods, making the explanation less convincing.
2. It is unclear whether the fluorescence of the analyte itself interacts with the enhanced fluorescence signal upon binding with the receptor. The text mentions, "silica core nanoparticles, which in turn contain a dye with a second fluorescence color as a reference signal." Does this dye color combine with the green fluorescence to form a mixed color, or is it merely a change in fluorescence intensity?
3. Although the molecular imprinting recognition layer and extraction steps aim to reduce interference from competitors or matrix effects, the synthesis process is complex with numerous influencing factors, making precise control of detection challenging.
4. The statement "The proton-assisted hydrogen bonding of the analyte significantly enhances fluorescence" requires further explanation.
5. The claim "These results indicate that 1 is less easily protonated by OA and that the hydrogen-bonded partners interact less strongly in the complex compared to 1 and PFOA" raises the question of whether other carboxylic acids can acidify 1 as effectively as PFOA.
6. The main text does not clearly present the calculation process and curves for combining constants and detection limits, nor the selection of detection ranges. This lack of clarity in the arrangement and explanation of data graphs can lead to misunderstandings.

Reviewer #3

(Remarks to the Author)

The authors introduce a rationally designed guanidine-BODIPY-based fluorescent probe combined with molecular imprinting polymer (MIP) for the selective detection of perfluorooctanoic acid (PFOA). The probe complexes PFOA through concerted protonation-induced ion pairing-assisted hydrogen bonding, showing a turn-on green fluorescence signal in MeCN. Then, the MIP was formed as a thin layer on silica nanoparticles, providing an orange emission signal as internal reference. Finally, the miniaturized microfluidic device can realize the convenient detection of PFOA in environmental water samples.

In this manuscript, the authors not only developed a PFOA sensor but also drew rich conclusions. For example, after obtaining response signals from interfering substances, the authors extensively discussed the mechanisms behind this phenomenon, which is highly commendable. Furthermore, encountering the contradiction between the high sensitivity of the MeCN system provided and the influence of MeCN-water co-solvent on precision during the research process, the authors proposed a biphasic ternary MeCN/EtOAc/H₂O solvent system to address this issue. The authors conducted a comprehensive comparison with many previous works in the Supplementary Information, providing an intuitive and objective demonstration of the advantages of this study. Publication of this article in Nature Communications is recommended following minor revision.

1. On page 11, line 386, when extraction and detection steps are separated, what is the extraction efficiency of PFOA from water samples using EtOAc, and will it cause the test results to be low?
2. In Figure S20a, please indicate the meaning of the black and red lines.
3. It is recommended to add photographs of the microfluidic device.
4. "Dual fluorescent" is mentioned multiple times in this manuscript, and the author should provide a detailed explanation of its advantages compared to other materials.
5. In the manuscript, the authors mention that the optimal flow rate of the solution in the opto-microfluidic platform is 11 $\mu\text{L min}^{-1}$, so was the platform built by the author themselves? How is the optimal flow rate determined?
6. The authors should explain the experimental details of liquid-liquid extraction and how this step avoids the adsorption loss of PFOA analyte during the extraction process?
7. The format of the horizontal axis in Figures 4b and c is inconsistent. It is recommended that the author make modifications.

Version 1:

Reviewer comments:

Reviewer #2

(Remarks to the Author)

[Note from the Editor: Reviewer #2 was asked to assess also the response given to reviewer #1.]

The authors have fully addressed my concerns, and the revised manuscript meets my expectations. I recommend acceptance pending minor editorial edits to (i) explicitly attribute the initial titration curvature to PFOA homoconjugation in MeCN and (ii) clarify that this effect is negligible under assay conditions where water is present. No additional experiments are required.

Reviewer #3

(Remarks to the Author)

The authors have addressed all my concern. I recommend to accept this manuscript.

General response:

We would like to thank the reviewers for their careful review and valuable comments as well as for their positive evaluation of our work. Before going into the detailed responses, we would like to point out the challenge of fully capturing the complexity of developing a supramolecular chemistry-based sensor system in a single manuscript. While a detailed examination of each aspect would be ideal, it is essential to balance the need for a comprehensive explanation with publication guidelines, particularly with regard to word limits (which we have already exceeded in the original version) and the number of references.

To address this challenge, we have placed most of the additional experimental data, information and discussion that we included in response to the reviewer comments during the revision in the Supplementary Information (SI). This approach allows us to provide the requested deeper insights into various aspects, such as host-guest chemistry, photophysics, and analytical metrics, without compromising the clarity and focus of the main text. In this way, the core message of the manuscript is preserved, namely the introduction of a novel, miniaturized detection platform for PFOA.

We believe that this compromise fully addresses the reviewers' comments while adhering to the journal's guidelines. However, we are open to moving certain aspects from the SI to the main text if the editor deems it necessary to improve the overall understanding and impact of the paper.

Specific responses:

Reviewer 1:

To my opinion, Rurack et al. has established an very interesting method to detect perfluoroalkyl carboxylic acids, a family of emerging pollutants whose concern is growing rapidly and, for this reason, there is a huge demand, even rightly coming from legislators, to check their concentrations in water also using simple approaches. Reading this article, I have been impressed by the high level of detail. The thorough explanations, comprehensive analysis, and meticulous attention to every aspect have yielded a high level article. The authors made a very extensive and accurate work reaching promising results, and for all these reasons I suggest its publication although not in the present form.

Reply:

We thank the reviewer for the encouraging assessment of our work and for the constructive comments. We have taken these comments into account in the revision, and we believe that especially the additional investigations and explanations in connection with Comment 1 highlight more clearly both the broader relevance of our findings and the novelty of our approach.

Comment 1:

I have in fact a major concern, that is related to the shape of the titration profiles such the one reported in the inset of figure 2b. A pure 1:1 stoichiometry, even with high association constants, to my knowledge, should not lead to an upward curvature in the first part of the titration, so that, if I am not wrong, this behaviour could hide another process that could be worth to investigate.

Reply 1:

That is an excellent observation, and we thank the reviewer for his/her careful inspection of our manuscript. The truth is that we have also noticed this unusual behaviour while working on this project, yet because the answer to the question is yes and no, because there is no published data available on several of the key factors that lead to the seemingly strange behaviour for the respective components of our system, and because the problem is not so easy to solve for an isolated system with a meaningful set of experiments, we had decided not to overload the present manuscript, but to address this issue in another study with better characterized species and other collaborators, which is

still ongoing. We have therefore included as much additional data and explanation as was necessary to support the consistency of the system despite the apparent contradictions and explain the issue here in brief. The results of a more general and comprehensive investigation will be published separately.

In a simplified manner, the two-fold yes/no answer is that the stoichiometry is 1:1 but that the guest (PFOA) is not only existing in a single form. The reason for this is the homoconjugation of acids in organic solvents. This is a known complication for electrochemical studies of acids in organic solvents (see, e.g., Chapter 3 "Acid-Base Reactions in Nonaqueous Solvents" in Izutsu K. *Electrochemistry in Nonaqueous Solutions*, 2 edn. Wiley-VCH (2009)), where for instance the adherence to strictly controlled and/or inert conditions is crucial but is neglected in most supramolecular studies using, for instance, NMR or optical spectroscopy. Also, the fact that many supramolecular anion complexes are only moderately strong contributes to the fact that this issue often escapes unnoticed.

Homoconjugation means that the acid is not only existing as neutral and deprotonated species (eq. 1, acidity constant K_a , or more precisely in an organic solvent, apparent acidity constant K_a^{app}), but that the acid also forms a dimeric species (eq. 2, homoconjugation constant K_{AHA^-}):

Depending on K_a^{app} and K_{AHA^-} , the number of free protons provided by one molecule of a certain acid under specific conditions can vary between 1 and 0.5, in the latter case for the overall reaction, eq. 3:

Such anionic species, as well as neutral dimeric species, have for instance recently been found in the gas phase by ion mobility spectrometry (IMS) and FTIR studies (e.g., *J Am Soc Mass Spectrom* **36**, 850 (2025); *Molecules* **30**, 1887 (2025)).

The situation is further complicated by the fact that not only homoconjugation can occur but that acids in organic solvents can also undergo heteroconjugation:

Here, HR is considered as an impurity in the system, yet is in most practical cases equal to water which is still present in traces especially if solvents and compounds have not been meticulously dried and if the system has not been investigated under inert conditions. The latter is naturally more important for inorganic acids.

If we neglect the heteroconjugation case and consider our present system, the homoconjugate and the free anion will certainly have a distinctly different affinity for a protonated guanidinium binding site, $[AHA]^-$ being most likely of a structure such as

In addition, the way the two species A^- (= PFOA⁻) and $[AHA]^-$ (= [PFOA-H-PFOA]⁻) can interact with the receptor unit of indicator **1** are distinctly different, which would presumably result in largely different binding constants and also different spectroscopic responses. However, the titration spectra

always show clear isosbestic points, suggesting the presence of only two species absorbing in the visible spectral range (Figure S10).

The problem in our case are the unknown values for K_a^{app} and K_{AHA-} of PFOA in MeCN. Moreover, because already the pK_a of PFOA in water is a matter of debate and values between -0.5 and 3.8 have been reported in the literature (e.g., *J Org Chem* **27**, 4491 (1962); *J Phys Chem* **89**, 5308 (1985); *Microchim Acta* **106**, 37 (1992); *Environ Sci Technol* **42**, 9283 (2008); *Environ Sci Technol* **42**, 456 (2008); *J Phys Chem A* **113**, 8152 (2009)), it is not trivial to obtain reliable data to quantitatively assess the different equilibria. Because except for neutral and protonated **1** all the other species escape optical detection and because the behaviour of the complicated system at low concentrations is not directly comparable to NMR concentrations, the question could not be directly solved. However, as we agree with the reviewer that the plot and data given in the manuscript can indeed be misleading, we included a new section into the text on p. 8–9 in which we outline the main reasons for the behaviour (p. 8, l. 5–p. 9, l. 21) and elaborated on the complexity of the case in Section VII (p. S23–S26) of the Supplementary Information.

First additional remark: The peculiarity of this initial phase in a titration curve vanishes from visibility when one uses a representation that shows more of the plateau region at higher analyte/guest concentrations/excess, as is often done in the literature. It is of course also much less visible for systems that have smaller complexation constants for which higher amounts of guest are necessary to induce effects and reach saturation, like in our case for OA as shown in Figure S8. A closer zoom into the low concentration range would show that the S-shaped onset is also visible for OA, as would be expected for a lipophilic acid such as OA which will most likely also show homoconjugation in MeCN.

Second additional remark: Although we were able to obtain a titration curve typical for a 1:1 stoichiometry for the system PFOA/**1**/MeCN, see Figures 2g and S9a, we could not achieve this for the acetate solvents within reasonable effort. Apparently, the equilibria are very much solvent-dependent. Nevertheless, to be able to compare the reproducible S-shaped response behaviour, we changed all the K_S data to apparent (K_S^{app}) data after reanalysing the titration curves according to a nonlinear dose–response behaviour, which is used e.g. in immunoanalysis when binders/receptors cannot be considered homogeneous (e.g., polyclonal antibodies), except that in our case the guest is the heterogeneous component of the system and not the host. We also validated all these experiments with LC/MS studies. However, LC/MS is only able to provide the total amount of PFOA, not the single species, and can thus only exclude changes in species concentration because of adsorption to container walls or other experimental effects.

Third additional remark: We carried out detailed time-resolved fluorescence experiments to probe the situation in more detail, as such measurements provide markedly higher discriminatory power by distinguishing between fluorescence-enhancing and quenching species that are otherwise spectrally indistinguishable.

The newly added experiments, data and texts can be found in Figure 2f–h (titrations, time-resolved fluorescence experiments) and on p. 8–9 (considerations on homoconjugation, titrations, time-resolved fluorescence experiments) of the main text as well as on p. S4–S5, including Tables S1–S3 (experimental details on time resolved fluorescence studies and LC/MS validation experiments), Section I.c.i (p. S6, additional titration experiments), Section I.f (p. S8, binding constant and apparent binding constant determination), Figure S9 (additional titration data), Section VI, including Table S9 issue (p. S21–S22, details on time-resolved fluorescence experiments) and Section VII, including Figures S10, S11 (p. S23–S26, considerations on titrations, species diversity and binding modes) in the Supplementary Information.

Concluding remark: Once a certain amount of water is present in the system, including for instance the extraction from an aqueous phase (because water and EtOAc are miscible to a minor extent, see ref. 63 in the main text), homoconjugation does not occur anymore, simplifying the system tremendously. These aspects are thus only important for model studies in organic solvents but do not affect the analytical assay.

Comment 2:

As a very minor point, I would also suggest not to stress – in the introduction – that previous approaches could have, as possible drawback, the use of “significant amounts of organic solvents” or, better, to make a lower use of organic solvents among the possible improvements of their approach mentioned in the last sentence of the discussion.

Reply 2:

We agree with the reviewer and toned this done by omitting “significant amounts” (p.2, bottom of first paragraph) yet stressing the advantage of our approach by adding a subclause to the last bullet point on p. 2 (vii) and a further point to the list on p. 18. The respective passages read now like:

However, these approaches typically require lab-based instrumentation and suffer from drawbacks such as complexity, e.g., in multi-component detection schemes involving micelle or aggregate formation, long assay times, use of toxic materials or organic solvents. (p. 2)

(vii) a simple phase-transfer shaking step, the microfluidic approach only requiring small amounts of an organic solvent. (p. 2)

Compared to existing technologies, our sensing system provides several distinct advantages such as direct indication via host-guest interaction, high selectivity to strong hydrophobic organic acids, effective matrix suppression, minimum sample treatment, fast analysis time and low reagent consumption, including organic solvents (Table S14). (p. 18)

Reviewer 2:

Rurack and coauthors reported synthesis of a novel BODIPY indicator monomer functionalized with green fluorescent guanidine, combined with molecularly imprinted polymers (MIPs) for the detection of PFOA in water samples via liquid-liquid extraction. While the mechanism behind the fluorescence enhancement is discussed, there are several limitations in this study. The research findings do not fully support the claims made and lack significant innovation, leading to the recommendation against publication in nature communications.

Reply:

We would like to thank the reviewer for his/her overall positive assessment of our work and for his/her constructive comments, which we are pleased to address below and which we hope will further stress the importance and novelty of our work.

Comment 3:

1. The discussion of the mechanism of fluorescence signal change following the interaction between the analyte and the receptor lacks supporting evidence from characterization methods, making the explanation less convincing.

Reply 3:

We apologize if the mechanistic description has been too short. In order to furnish additional proof and support, we have included the fluorescence lifetime data on **1**, **1**-PFOA and **1**/HCl in MeCN (Table S9, Section VI, Supplementary Information) and discussed the trends in fluorescence quantum yields for various species (p. 7–8). In addition, we added two new paragraphs on p. 5, 7 and new panels to Figure 2 as well as a new Section IV, including Figure S5 and Tables S5–S7, in the Supplementary Information to better illustrate the design considerations with the aid of theoretical results and additional experimental data. Such considerations and theoretical studies are commonly the starting point of our research projects. Figure 1 was also modified to illustrate the underlying rationale. In addition, the NMR experiments reported in Section VIII, Supplementary Information furnish proof by an independent method, the entire package of optical and NMR spectroscopy plus theoretical studies providing comprehensive evidence of the supramolecular signalling mechanism. Because of the peculiarities of PFOA in organic solvents as discussed in detail in the Sections included in response to Comment 1 of reviewer 1, we refrained from other additional experimental investigations.

The newly added experiments, data and texts can be found in Figure 1a,b (mechanistic considerations), Figure 2a–c, f–h (quantum chemical calculations, time-resolved fluorescence experiments), p. 5 (paragr. 3, mechanistic considerations, quantum chemical calculations) and on p. 7–8 (mechanistic studies) of the main text, the latter including the investigation of two newly included model compounds **4** and **5** (molecular structures in Table S8), as well as on p. S4 (experimental details on fluorescence studies), Section I.c.i (p. S6, additional titration experiments), Section I.f (p. S8, binding constant and apparent binding constant determination), Section IV, including Tables S5–S7 and Figure S5 (p. S14–S18, theoretical considerations on signalling mechanism), Table S8 (spectroscopic and photophysical data), Figure S9 (additional titration data) and Section VI, including Table S9 (p. S21–S22, time-resolved fluorescence studies of host-guest interaction).

Comment 4:

2. It is unclear whether the fluorescence of the analyte itself interacts with the enhanced fluorescence signal upon binding with the receptor. The text mentions, "silica core nanoparticles, which in turn contain a dye with a second fluorescence color as a reference signal." Does this dye color combine with the green fluorescence to form a mixed color, or is it merely a change in fluorescence intensity?

Reply 4:

We thank the reviewer for pointing out that our explanation might be misunderstood and have added/modified some parts of the text to make it clearer.

The analyte itself, PFOA, is essentially non-fluorescent, does not even absorb in the optically transparent range of MeCN, i.e., ≥ 250 nm, see absorption spectrum of a 2 mM PFOA solution in MeCN below. There is thus no photophysical interaction between PFOA and indicator **1**. The fluorescence enhancement upon binding of PFOA to **1** is therefore solely due to the two processes described in the text. First, **1** is protonated by PFOA, turning the weakly emissive **1** into the highly emissive **1H⁺**. The weak emission is due to an active electron transfer process as described in the new passages on p. 5, 7 in the text and Section IV, p. S14–S18, Supplementary Information, and the switching on by protonation follows the same mechanism as has been previously reported for main group metal ions and azacrown receptors (*J Phys Chem A* **102**, 10211 (1998)), heavy and transition metal ions and azathiacrowns (*J Am Chem Soc* **122**, 968 (2000)) or aniline-based BODIPY pH indicators (*Anal Chem* **89**, 8437 (2017)).

Absorption spectrum of 2 mM PFOA in MeCN.

With respect to the fluorescent silica core nanoparticles, these contain a Ru(bipy)₃ dye that is absorbing at shorter and emitting at longer wavelengths than the BODIPY indicator monomer. Thus, as shown in Figure 3a and Figures S22, S23, the sensory particles contain two different fluorescent species in their two different compartments, core and shell, which are separated and do not interact chemically or photophysically, see the spectroscopic analysis for potential energy transfer interactions on p. S37, Supplementary Information.

The visual appearance of a suspension of the particles would essentially be a mixed colour, i.e., a mixture of the complementary colour of the absorbed light and the two emission colours, but a spectrometer or an adequately equipped simpler measurement setup like the one used here in combination with the microfluidic chip allow to spectrally separate the analyte-sensitive green and the constant red (reference) signal, despite that there is a certain overlap, see Figure S25. Essentially, as explained various times in the text, the intensity of the green contribution varies with analyte concentration whereas the red contribution remains constant. Because the absorption of **1** only shifts by a few nanometres upon binding PFOA, the colour appearance of a suspension of the particles mainly varies with the amount of green fluorescence from bound **1** and changes, figuratively speaking, from a reddish orange to a brownish beige.

Comment 5:

3. Although the molecular imprinting recognition layer and extraction steps aim to reduce interference from competitors or matrix effects, the synthesis process is complex with numerous influencing factors, making precise control of detection challenging.

Reply 5:

This observation by the reviewer is certainly valid, however, to the best of our knowledge no indicator that by itself can selectively bind and detect PFOA, PFOS, PFCAs or PFAAs has been reported yet. Synthesizing a chemical receptor with such properties would certainly require a multistep organic synthesis, if being possible at all. Raising antibodies or other biological binders against these analytes has also not been satisfactorily successful yet. Addressing PFOA *et al.* by chemical recognition is a challenge. However, we are confident that our approach is a good compromise between performance, effort and cost.

As shown in Figure 3a, the synthesis route is rather simple and short. The reference dye is a commercial compound which is simply sterically embedded into the core particle during its synthesis. Functionalization is then also a straightforward two-step process, as is the growth of the final polymer shell. The synthesis of the BODIPY indicator monomer is also not too challenging, being a green, *meso*-substituted BODIPY which can be readily synthesized in acceptable yields. Moreover, as we have recently shown for another dual fluorescent core-shell system, which is even more complex as it follows a core-shell-shell architecture, the batch-to-batch reproducibility of such systems is very good (*ACS Appl Mater Interfaces* **16**, 49944 (2024)). Furthermore, in a detailed study of other core-shell-shell particles to which biological binders are attached in different ways, we have recently shown that the synthesis is highly reproducible, and that the performance of such systems is essentially determined by the outermost, analyte-sensitive layer (*ACS Meas Sci Au* **4**, 678 (2024)). As our protocols for the growth of an outer MIP shell have been refined and established over many years, we expect the systems to be highly reproducible and are confident that our approach, which is generic in itself, is an excellent combination of a well-defined architecture, facile synthesis and powerful response process.

Comment 6:

4. The statement "The proton-assisted hydrogen bonding of the analyte significantly enhances fluorescence" requires further explanation.

Reply 6:

We thank the reviewer for this comment and, in conjunction with what we replied on Comment 3 above, included a new Section IV in the Supplementary Information (p. S14–S18). In addition, we made several other small modifications in the text, see also Reply 3 above, that describe and illustrate the design considerations better.

Comment 7:

5. The claim "These results indicate that **1** is less easily protonated by OA and that the hydrogen-bonded partners interact less strongly in the complex compared to **1** and PFOA" raises the question of whether other carboxylic acids can acidify **1** as effectively as PFOA.

Reply 7:

As is written in the text, indicator **1** itself cannot discriminate between sufficiently strong acids. It can also not discriminate between different carboxylic acids solely based on chemical substitution patterns. Therefore, indicator **1** binds to HCl and would also bind to, for instance, perfluorohexanoic acid. As written in Reply 5 above, trying to achieve this selectivity exclusively with an indicator molecule is virtually impossible or would be a tremendous synthetic challenge. The discrimination can only be accomplished if all the parameters discussed in the context of Table S13 and Figure S32 are

considered. However, molecular discrimination would not be a problem from an application-oriented point of view for such a rapid testing method, because the end user would commonly not be interested in only a single specific PFCA or PFSA, but in the sum of medium- and long-chain PFCAs and PFSA. This is what our system can accomplish.

Comment 8:

6. The main text does not clearly present the calculation process and curves for combining constants and detection limits, nor the selection of detection ranges. This lack of clarity in the arrangement and explanation of data graphs can lead to misunderstandings.

Reply 8:

We apologize if the description and representation of these values lacked clarity and have added references to explanations in the *Methods* section, subsection *Signal processing* (p. 20) as well as in Sections I.f. and I.g., Supplementary Information (p. S8) and in the main text on p. 12, 13. As can be seen from Eqs. 4–8 and Eqs. S4–S6, F always corresponds to an integrated fluorescence intensity, S to a ratio of two fluorescence intensities, ΔF and ΔS to the difference between the intensity (or intensity ratio) in the presence of an analyte/species at a certain concentration and the intensity (or intensity ratio) in the absence of the analyte/species, and the reduced fluorescence (or fluorescence ratio) to the normalized signal changes. Because both, the fluorometer and the USB spectrometer used for the microfluidics are operated in photon counting modii, the signal handling is consistent in itself.

The most important equations for measurement uncertainties and limit of detection are also included in the *Methods* section of the main text (p. 20–21) and in the Supplementary Information, Sections I.d and I.e. The equations used for the fitting of the titration data are only included in the Supplementary Information, Sections I.f and I.g (p. S8), because the required explanation would otherwise unnecessarily lengthen the text. In conclusion, the *Signal processing* subsection in *Methods* now contains all the essential descriptions and equations.

Reviewer 3:

The authors introduce a rationally designed guanidine-BODIPY-based fluorescent indicator combined with molecular imprinting polymer (MIP) for the selective detection of perfluorooctanoic acid (PFOA). The indicator complexes PFOA through concerted protonation-induced ion pairing-assisted hydrogen bonding, showing a turn-on green fluorescence signal in MeCN. Then, the MIP was formed as a thin layer on silica nanoparticles, providing an orange emission signal as internal reference. Finally, the miniaturized microfluidic device can realize the convenient detection of PFOA in environmental water samples.

In this manuscript, the authors not only developed a PFOA sensor but also drew rich conclusions. For example, after obtaining response signals from interfering substances, the authors extensively discussed the mechanisms behind this phenomenon, which is highly commendable. Furthermore, encountering the contradiction between the high sensitivity of the MeCN system provided and the influence of MeCN-water co-solvent on precision during the research process, the authors proposed a biphasic ternary MeCN/EtOAc/H₂O solvent system to address this issue. The authors conducted a comprehensive comparison with many previous works in the Supplementary Information, providing an intuitive and objective demonstration of the advantages of this study. Publication of this article in Nature Communications is recommended following minor revision.

Reply:

We are grateful to the reviewer for the careful reading of our manuscript and for the constructive feedback provided. We have considered all points in detail, as outlined below, and we are convinced that the revisions improve the overall quality and impact of the work.

Comment 9:

1. On page 11, line 386, when extraction and detection steps are separated, what is the extraction efficiency of PFOA from water samples using EtOAc, and will it cause the test results to be low?

Reply 9:

We thank the reviewer for this comment and have included the relevant data in a footnote to Figure S29 in the SI, namely the EtOAc-water partition coefficient of PFOA. It should also be noted that the acidification step to pH 2 in the assay fulfils two roles, converting PFOA⁻ anions into the extractable neutral PFOA form and thereby increases K_{EW} . Together with another research group that has specialized in combustion ion chromatography (CIC) we have studied the partition behaviour of various PFCAs and will report on these results in a separate paper in due course.

Figure S34 shows the absence of memory effects. The latter is expected to be negligible as PFOA is well soluble in the ternary solvent mixture injected in the chip, minimizing adsorption/desorption to/from tubing in the microfluidic setup.

Comment 10:

2. In Figure S20a, please indicate the meaning of the black and red lines.

Reply 10:

Thank you for spotting the omission in the description of the colour code. The codes were added in the caption of (now) Figure S25.

Comment 11:

3. It is recommended to add photographs of the microfluidic device.

Reply 11:

We followed the recommendation of the reviewer and included a photograph of the lab prototype as Figure S32 in the Supplementary Information, mentioned in the text in the second paragraph of p. 17. As we commonly use pumps, excitation sources and detectors in various projects, these components are not specifically miniaturized or adapted for the current setup. A final device would naturally look different and more integrated.

Comment 12:

4. "Dual fluorescent" is mentioned multiple times in this manuscript, and the author should provide a detailed explanation of its advantages compared to other materials.

Reply 12:

We thank the reviewer for this suggestion and have expanded and reformatted the first paragraph of the Results section on p. 4, including the major reasons in bullet point (ii) and adding two authoritative review articles as new refs 42, 43 to which, in the interest of article length, we refer the reader for a more comprehensive account on such systems. The immediate advantages that become obvious from the results we report should be seen in light of this paragraph on system design.

Comment 13:

5. In the manuscript, the authors mention that the optimal flow rate of the solution in the opto-microfluidic platform is $11 \mu\text{L min}^{-1}$, so was the platform built by the author themselves? How is the optimal flow rate determined.

Reply 13:

For this work, the setup was built by us (new Figure S32) yet we used a commercial pearl chain mixer chip from ChipShop Jena this time, as written on p. 17, *Opto-microfluidic platform ...*. The mixing efficiency was tested with dyed solutions, revealing no significant differences between $1\text{--}40 \mu\text{L min}^{-1}$ (new Figure S33) and the medium flow rate of $11 \mu\text{L min}^{-1}$ was chosen to guarantee stable flows and signals within the entire setup. The flow rate is less critical in this work, because we opted for a separate extraction step. If the extraction step should be integrated into the microfluidic device, extensive optimization procedures as recently described by us in *Lab Chip* **23**, 466-474 (2023) would be required. Because the focus of this work was not primarily on the device side, but on the sensory particles, the device was conceived to guarantee a residence time of the sample in the setup of 10 min (Section XII.b, p. S45). The respective additions have been made in the microfluidics section (Section XII) of the Supplementary Information, p. S44–S45.

Comment 14:

6. The authors should explain the experimental details of liquid-liquid extraction and how this step avoids the adsorption loss of PFOA analyte during the extraction process.

Reply 14:

Please see our Reply to Comment 9 above.

Comment 15:

7. The format of the horizontal axis in Figures 4b and c is inconsistent. It is recommended that the author make modifications.

Reply 15:

Thank you for spotting this issue, which we corrected.